# NEO-KD: Knowledge-Distillation-Based Adversarial Training for Robust Multi-Exit Neural Networks

**Seokil Ham**[1]    **Jungwuk Park**[1]    **Dong-Jun Han**[2]*    **Jaekyun Moon**[1]
[1]KAIST    [2]Purdue University
{gkatjrdlf, savertm}@kaist.ac.kr, han762@purdue.edu, jmoon@kaist.edu

## Abstract

While multi-exit neural networks are regarded as a promising solution for making efficient inference via early exits, combating adversarial attacks remains a challenging problem. In multi-exit networks, due to the high dependency among different submodels, an adversarial example targeting a specific exit not only degrades the performance of the target exit but also reduces the performance of all other exits concurrently. This makes multi-exit networks highly vulnerable to simple adversarial attacks. In this paper, we propose NEO-KD, a knowledge-distillation-based adversarial training strategy that tackles this fundamental challenge based on two key contributions. NEO-KD first resorts to neighbor knowledge distillation to guide the output of the adversarial examples to tend to the ensemble outputs of neighbor exits of clean data. NEO-KD also employs exit-wise orthogonal knowledge distillation for reducing adversarial transferability across different submodels. The result is a significantly improved robustness against adversarial attacks. Experimental results on various datasets/models show that our method achieves the best adversarial accuracy with reduced computation budgets, compared to the baselines relying on existing adversarial training or knowledge distillation techniques for multi-exit networks.

## 1   Introduction

Multi-exit neural networks are receiving significant attention [9, 13, 26, 27, 28, 32] for their ability to make dynamic predictions in resource-constrained applications. Instead of making predictions at the final output of the full model, a faster prediction can be made at an earlier exit depending on the current time budget or computing budget. In this sense, a multi-exit network can be viewed as an architecture having multiple submodels, where each submodel consists of parameters from the input of the model to the output of a specific exit. These submodels are highly correlated as they share some model parameters. It is also well-known that the performance of all submodels can be improved by distilling the knowledge of the last exit to other exits, i.e., via self-distillation [15, 20, 24, 27]. There have also been efforts to address the adversarial attack issues in the context of multi-exit networks [3, 12].

Providing robustness against adversarial attacks is especially challenging in multi-exit networks: since different submodels have high correlations by sharing parameters, an adversarial example targeting a specific exit can significantly degrade the performance of other submodels. In other words, an adversarial example can have strong *adversarial transferability* across different submodels, making the model highly vulnerable to simple adversarial attacks (e.g., an adversarial attack targeting a single exit).

---

*Corresponding author.

37th Conference on Neural Information Processing Systems (NeurIPS 2023).

**Motivation.** Only a few prior works have focused on adversarial defense strategies for multi-exit networks [3, 12]. The authors of [12] focused on generating adversarial examples tailored to multi-exit networks (e.g., generate samples via max-average attack), and trained the model to minimize the sum of clean and adversarial losses of all exits. Given the adversarial example constructed in [12], the authors of [3] proposed a regularization term to reduce the weights of the classifier at each exit during training. However, existing adversarial defense strategies [3, 12] do not directly handle the high correlations among different submodels, resulting in high adversarial transferability and limited robustness in multi-exit networks. To tackle this difficulty, we take a knowledge-distillation-based approach in a fashion orthogonal to prior works [3, 12]. Some previous studies [8, 23, 33, 34] have shown that knowledge distillation can be utilized for improving the robustness of the model in conventional single-exit networks. However, although there are extensive existing works on self-distillation for training multi-exit networks using clean data [15, 20, 24, 27], it is currently unknown how distillation techniques should be utilized for adversarial training of multi-exit networks. Moreover, when the existing distillation-based schemes are applied to multi-exit networks, the dependencies among submodels become higher since the same output (e.g., the knowledge of the last exit) is distilled to all sub-models. Motivated by these limitations, we pose the following questions: *How can we take advantage of knowledge-distillation to improve adversarial robustness of multi-exit networks? At the same time, how can we reduce adversarial transferability across different submodels in multi-exit networks?*

**Main contributions.** To handle these questions, we propose NEO-KD, a knowledge-distillation-based adversarial training strategy highly tailored to robust multi-exit neural networks. Our solution is two-pronged: neighbor knowledge distillation and exit-wise orthogonal knowledge distillation.

- Given a specific exit, the first part of our solution, neighbor knowledge distillation (NKD), distills the ensembled prediction of neighbor exits of clean data to the prediction of the adversarial example at the corresponding exit, as shown in Figure 1a. This method guides the output of adversarial examples to follow the outputs of clean data, improving robustness against adversarial attacks. By ensembling the neighbor predictions of clean data before distillation, NKD provides higher quality features to the corresponding exits compared to the scheme distilling with only one exit in the same position.

- The second focus of our solution, exit-wise orthogonal knowledge distillation (EOKD), mainly aims at reducing adversarial transferability across different submodels. This part is another unique contribution of our work compared to existing methods on robust multi-exit networks [3, 12] (that suffer from high adversarial transferability) or self-distillation-based multi-exit networks [15, 20, 24, 27] (that further increase adversarial transferability). In our EOKD, the output of clean data at the $i$-th exit is distilled to the output of the adversarial sample at the $i$-th exit, in an exit-wise manner. During this exit-wise distillation process, we encourage the non-ground-truth predictions of individual exits to be mutually orthogonal, by providing orthogonal soft labels to each exit as described in Figure 1b. By weakening the dependencies among different exit outputs, EOKD reduces the adversarial transferability across all submodels in the network, which leads to an improved robustness against adversarial attacks.

The NKD and EOKD components of our architectural solution work together to reduce adversarial transferability across different submodels in the network while correctly guiding the predictions of the adversarial examples at each exit. Experimental results on various datasets show that the proposed strategy achieves the best adversarial accuracy with reduced computation budgets, compared to existing adversarial training methods for multi-exit networks. Our solution is a plug-and-play method, which can be used in conjunction with existing training strategies tailored to multi-exit networks.

## 2 Related Works

**Knowledge distillation for multi-exit networks.** Multi-exit neural networks [9, 13, 26, 27, 28, 32] aim at making efficient inference via early exits in resource-constrained applications. In the multi-exit network literature, it is well-known that distilling the knowledge of the last exit to others significantly improves the overall performance on clean data without an external teacher network, i.e., via self-distillation [15, 20, 24, 27]. However, it is currently unclear how adversarial training can benefit from self-distillation in multi-exit networks. One challenge is that simply applying existing self-distillation techniques increases adversarial transferability across different submodels, since the same knowledge

from the last exit is distilled to all other exits, increasing dependency among different submodels in the network. Compared to the existing ideas, our contribution is to develop a self-distillation strategy that does not increase the dependency of submodels as much; this helps reduce adversarial transferability of the multi-exit network for better robustness.

**Improving adversarial robustness.** Most existing defense methods [6, 7, 31] have mainly focused on creating new adversarial training losses tailored to single-exit networks. Several other works have utilized the concept of knowledge distillation [8, 23, 33, 34] showing that distilling the knowledge of the teacher network can improve robustness of the student network. Especially in [8], given a teacher network, it is shown that robustness of the teacher network can be distilled to the student network during adversarial training. Compared to these works, our approach can be viewed as a new self-distillation strategy for multi-exit networks where teacher/student models are trained together. More importantly, adversarial transferability across different submodels has not been an issue in previous works as the focus there has been on the single-exit network. In contrast, in our multi-exit setup, all submodels sharing some model parameters require extra robustness against adversarial attacks; this motivates us to propose exit-wise orthogonal knowledge distillation, to reduce adversarial transferability among different submodels.

Some prior works [19, 22, 29, 30] aim at improving adversarial robustness of the ensemble model, by reducing adversarial transferability across individual models. Specifically, the adaptive diversity-promoting regularizer proposed in [22] regularizes the non-maximal predictions of individual models to be mutually orthogonal, and the maximal term is used to compute the loss as usual. While the previous work focuses on reducing the transferability among different models having independent parameters, in a multi-exit network setup, the problem becomes more challenging in that all submodels have some shared parameters, making the models to be highly correlated. To handle this issue, we specifically take advantage of knowledge distillation in an exit-wise manner, which can further reduce the dependency among different submodels in the multi-exit network.

**Adversarial training for multi-exit networks.** When focused on multi-exit networks, only a few prior works considered the adversarial attack issue in the literature [3, 10, 11, 12]. The authors of [10, 11] focused on generating slowdown attacks in multi-exit networks rather than defense strategies. In [12], the authors proposed an adversarial training strategy by generating adversarial examples targeting a specific exit (single attack) or multiple exits (average attack and max-average attack). However, (i) [12] does not take advantage of knowledge distillation during training and (ii) [12] does not directly handle the high correlations among different submodels, which can result in high adversarial transferability. Our solution overcomes these limitations by reducing adversarial transferability while correctly guiding the predictions of adversarial examples at each exit, via self knowledge distillation.

## 3 Proposed NEO-KD Algorithm

Consider a multi-exit network with $L$ exits, which is composed of $L$ blocks $\{\phi_i\}_{i=1}^{L}$ and $L$ classifiers $\{w_i\}_{i=1}^{L}$. Given the input data $x$, the output of the $i$-th exit is denoted as $f_{\theta_i}(x)$, which is parameterized by the $i$-th submodel $\theta_i = [\phi_1, \ldots, \phi_i, w_i]$ that consists of $i$ blocks and one classifier. Note that all $L$ submodels produce different predictions $[f_{\theta_1}(x), \ldots, f_{\theta_L}(x)]$. Here, since each submodel shares several blocks with other submodels, the predictions of any two submodels are highly correlated.

### 3.1 Problem Setup: Adversarial Training in Multi-Exit Networks

The first step for adversarial training is to generate adversarial examples. Given $L$ different submodels $\{\theta_i\}_{i=1}^{L}$, clean data $x$, and the corresponding label $y$, the adversarial example $x^{adv}$ can be generated based on single attack, max-average attack or average attack, following the process of [12]. More specifically, we have

$$x_{single,i}^{adv} = \underset{x' \in \{z:|z-x|_\infty \leq \epsilon\}}{\arg\max} |\ell(f_{\theta_i}(x'), y)|, \tag{1}$$

$$x_{max}^{adv} = x_{i^*}^{adv}, \quad \text{where} \quad i^* = \underset{i}{\arg\max} \left| \frac{1}{L} \sum_{j=1}^{L} \ell(f_{\theta_j}(x_{single,i}^{adv}), y) \right|, \tag{2}$$

$$x_{avg}^{adv} = \underset{x' \in \{z:|z-x|_\infty \leq \epsilon\}}{\arg\max} \left| \frac{1}{L} \sum_{j=1}^{L} \ell(f_{\theta_j}(x'), y) \right|, \tag{3}$$

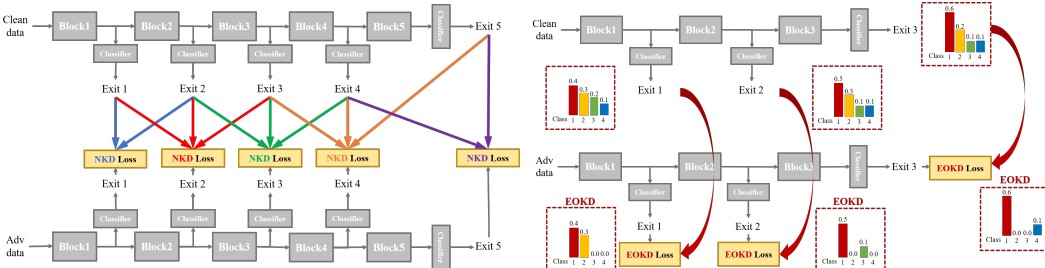

(a) Neighbor knowledge distillation  (b) Exit-wise orthogonal knowledge distillation

Figure 1: NEO-KD consists of two parts that together improve the adversarial robustness: NKD and EOKD. (a) NKD guides the output of the adversarial data to mimic the ensemble outputs of neighbor exits of clean data. (b) EOKD reduces adversarial transferability of the network by distilling orthogonal knowledge of the clean data to adversarial data for the non-ground-truth predictions, in an exit-wise manner. Although omitted in this figure, EOKD normalizes the likelihood before distilling the soft labels. The overall process operates in a single model, although we consider two cases depending on the input (clean or adversarial example) for a clear presentation.

which correspond to the adversarial example generated by single attack targeting exit $i$, max-average attack, and average attack, respectively. $\epsilon$ denotes the perturbation degree. In the single attack of (1), the adversarial example $x_i^{adv}$ is generated to maximize the cross-entropy loss $\ell(\cdot, \cdot)$ of the target exit utilizing an *attacker algorithm* (e.g., PGD [21]). In the max-average attack of (2), among the adversarial examples generated by the single attack for all exits $i = 1, 2, \ldots, L$, the sample that maximizes the average loss of all exits is selected. Finally, the average attack in (3) directly generates an adversarial sample that maximizes the average loss of all exits. Based on the generated $x^{adv}$, a typical strategy is to update the model considering both the clean and adversarial losses of all exits as follows:

$$\mathcal{L} = \frac{1}{N} \sum_{j=1}^{N} \sum_{i=1}^{L} [\ell(f_{\theta_i}(x_j), y_j) + \ell(f_{\theta_i}(x_j^{adv}), y_j)], \tag{4}$$

where $N$ is the number of samples in the training set and $x_j^{adv}$ is the adversarial example corresponding to clean sample $x_j$ generated by one of the attacks described above. However, the loss in (4) does not directly consider the correlation among submodels, which can potentially increase adversarial transferability of the multi-exit network.

## 3.2 Algorithm Description

To tackle the limitations of prior works [3, 12], we propose neighbor exit-wise orthogonal knowledge distillation (NEO-KD), a self-distillation strategy tailored to robust multi-exit networks. To gain insights, we divide our solution into two distinct components with different roles - neighbor knowledge distillation and exit-wise orthogonal knowledge distillation - and first describe each component separately, and then put together into the overall NEO-KD method. A high-level description of our approach is given in Figure 1.

**Neighbor knowledge distillation (NKD).** The first component of our solution, NKD, guides the output feature of adversarial data at each exit to mimic the output feature of clean data. Specifically, the proposed NKD loss of the $j$-th train sample at the $i$-th exit is written as follows:

$$NKD_{i,j} = \begin{cases} \ell(f_{\theta_1}(x_j^{adv}), \frac{1}{2} \sum_{k=1}^{2} f_{\theta_k}(x_j)) & i = 1 \\ \ell(f_{\theta_L}(x_j^{adv}), \frac{1}{2} \sum_{k=L-1}^{L} f_{\theta_k}(x_j)) & i = L \\ \ell(f_{\theta_i}(x_j^{adv}), \frac{1}{3} \sum_{k=i-1}^{i+1} f_{\theta_k}(x_j)) & \text{otherwise,} \end{cases} \tag{5}$$

which can be visualized with the colored arrows as in Figure 1a. Different from previous self-knowledge distillation methods, for each exit $i$, NKD generates a teacher prediction by ensembling (averaging) the neighbor predictions (i.e., from exit $i-1$ and exit $i+1$) of clean data and distills it to each prediction of adversarial examples.

Compared to other distillation strategies, NKD takes advantage of only these *neighbors* during distillation, which has the following key advantages for improving adversarial robustness. First,

by ensembling the neighbor predictions of clean data before distillation, NKD provides a higher quality feature of the original data to the corresponding exit; compared to the naive baseline that is distilled with only one exit in the same position (without ensembling), NKD achieves better adversarial accuracy, where the results are provided in Table 7. Secondly, by considering only the neighbors during ensembling, we can distill different teacher predictions to each exit. Different teacher predictions of NKD also play a role of reducing adversarial transferability compared to the strategies that distill the same prediction (e.g., the last exit) to all exit; ensembling other exits (beyond the neighbors) increases the dependencies among submodels, resulting in higher adversarial transferability. The corresponding results are also shown via experiments in Section 4.3.

However, a multi-exit network trained with only NKD loss still has a significant room for mitigating adversarial transferability further. In the following, we describe the second part of our solution that solely focuses on reducing adversarial transferability of multi-exit networks.

**Exit-wise orthogonal knowledge distillation (EOKD).** EOKD provides orthogonal soft labels to each exit for the non-ground-truth predictions, in an exit-wise manner. As can be seen from the red arrows in Figure 1b, the output of clean data at the $i$-th exit is distilled to the output of adversarial example at the $i$-th exit. During this exit-wise distillation process, some predictions are discarded to encourage the non-ground-truth predictions of individual exits to be mutually orthogonal. We randomly allocate the classes of non-ground-truth predictions to each exit for every epoch, which prevents the classifier to be biased compared to the fixed allocation strategy. The proposed EOKD loss of the $j$-th sample at the $i$-th exit is defined as follows:

$$EOKD_{i,j} = \ell(f_{\theta_i}(x_j^{adv}), O(f_{\theta_i}(x_j))). \tag{6}$$

Here, $O(\cdot)$ is the orthogonal labeling operation to make the non-ground-truth predictions to be orthogonal across all exits. For each exit, $O(\cdot)$ randomly selects $\lfloor (C-1)/L \rfloor$ labels among a total of $C$ classes so that the selected labels are non-overlapping across different exits (except for the ground-truth label), as in Figure 1b. Lastly, the probabilities of selected labels are normalized to sum to one. To gain clearer insights, consider a toy example with a 3-exit network (i.e., $L = 3$) focusing on a 4-way classification task (i.e., $C = 4$). Let $[p_1^i, p_2^i, p_3^i, p_4^i]$ be the softmax output of the clean sample at the $i$-th exit, for $i = 1, 2, 3$. If class 1 is the ground-truth, the orthogonal labeling operation $O(\cdot)$ jointly produces the following results from each exit: $[\hat{p}_1^1, \hat{p}_2^1, 0, 0]$ from exit 1, $[\hat{p}_1^2, 0, \hat{p}_3^2, 0]$ from exit 2, $[\hat{p}_1^3, 0, 0, \hat{p}_4^3]$ from exit 3, where $\hat{p}$ indicates the normalized probability of $p$ so that the values in each vector sum to one.

Based on Eq. (6), the output of the orthogonal label operation $O(f_{\theta_i}(x_j))$ for the clean data $x_j$ is distilled to $f_{\theta_i}(x_j^{adv})$ which is the prediction of the adversarial example of the $j$-th sample at the $i$-th exit. This encourages the model to self-distill orthogonally distinct knowledge in an exit-wise manner while keeping the essential knowledge of the ground-truth class. By taking this exit-wise orthogonal distillation approach, EOKD reduces the dependency among different submodels, reducing the adversarial transferability of the network.

**Overall NEO-KD loss.** Finally, considering the proposed loss functions in Eq. (5), (6) and the original adversarial training loss, the overall objective function of our scheme is written as follows:

$$\mathcal{L} = \frac{1}{N} \sum_{j=1}^{N} \sum_{i=1}^{L} [\ell(f_{\theta_i}(x_j), y_j) + \ell(f_{\theta_i}(x_j^{adv}), y_j) + \gamma_i(\alpha \cdot NKD_{i,j} + \beta \cdot EOKD_{i,j})], \quad (7)$$

where $\alpha, \beta$ control the weights for each component and $\gamma_i$ is the knowledge distillation weight for each exit $i$. Since later exits have lower knowledge distillation loss than the earlier exits, we impose a slightly higher $\gamma_i$ for the later exits than $\gamma_i$ of the earlier exits. More details regarding hyperparameters are described in Appendix.

By introducing two unique components - NKD and EOKD - the overall NEO-KD loss function in Eq. (7) reduces adversarial transferability in the multi-exit network while correctly guiding the output of the adversarial examples in each exit, significantly improving the adversarial robustness of multi-exit networks, as we will see in the next section.

## 4   Experiments

In this section, we evaluate our method on five datasets commonly adopted in multi-exit networks: MNIST [18], CIFAR-10, CIFAR-100 [16], Tiny-ImageNet [17], and ImageNet [25]. For MNIST, we

| Exit | 1 | 2 | 3 | Average | Exit | 1 | 2 | 3 | Average |
|---|---|---|---|---|---|---|---|---|---|
| Adv. w/o Distill [12] | 89.74% | 95.89% | 96.82% | 94.15% | Adv. w/o Distill [12] | 85.17% | 95.87% | 97.20% | 92.75% |
| SKD [24] | 89.77% | 96.24% | **97.08%** | 94.36% | SKD [24] | 84.35% | 96.53% | **97.53%** | 92.82% |
| ARD [8] | 89.65% | 95.79% | 96.47% | 93.97% | ARD [8] | 85.07% | 96.12% | 97.28% | 92.82% |
| LW [3] | 87.08% | 93.86% | 95.42% | 92.12% | LW [3] | 85.26% | 94.54% | 96.05% | 91.95% |
| NEO-KD (ours) | **90.56%** | **96.30%** | 96.62% | **94.49%** | NEO-KD (ours) | **86.17%** | **96.92%** | 97.42% | **93.50%** |

(a) Max-average attack  (b) Average attack

Table 1: **Anytime prediction setup**: Adversarial test accuracy on MNIST.

| Exit | 1 | 2 | 3 | Average | Exit | 1 | 2 | 3 | Average |
|---|---|---|---|---|---|---|---|---|---|
| Adv. w/o Distill [12] | 44.77% | 46.10% | 46.86% | 45.91% | Adv. w/o Distill [12] | 38.00% | 41.42% | 40.70% | 40.04% |
| SKD [24] | 44.75% | 44.54% | 44.79% | 44.69% | SKD [24] | 39.36% | 41.39% | 38.39% | 39.71% |
| ARD [8] | 44.50% | 45.85% | **51.82%** | 47.39% | ARD [8] | 39.37% | 41.98% | 43.53% | 41.63% |
| LW [3] | 37.38% | 35.39% | 34.85% | 35.87% | LW [3] | 31.47% | 31.41% | 28.98% | 30.62% |
| NEO-KD (ours) | **46.53%** | **47.65%** | 50.71% | **48.30%** | NEO-KD (ours) | **41.67%** | **45.38%** | **45.54%** | **44.20%** |

(a) Max-average attack  (b) Average attack

Table 2: **Anytime prediction setup**: Adversarial test accuracy on CIFAR-10.

use SmallCNN [12] with 3 exits. We trained the MSDNet [13] with 3 and 7 exits using CIFAR-10 and CIFAR-100, respectively. For Tiny-ImageNet and ImageNet, we trained the MSDNet with 5 exits. More implementation details are provided in Appendix.

## 4.1 Experimental Setup

**Generating adversarial examples.** To generate adversarial examples during training and testing, we utilize max-average attack and average attack proposed in [12]. We perform adversarial training using adversarial examples generated by max-average attack, where the results for adversarial training via average attack are reported in Appendix. During training, we use PGD attack [21] with 7 steps as attacker algorithm to generate max-average and average attack while PGD attack with 50 steps is adopted at test time for measuring robustness against a stronger attack. We further consider other strong attacks in Section 4.3. In each attacker algorithm, the perturbation degree $\epsilon$ is 0.3 for MNIST, and $8/255$ for CIFAR-10/100, and $2/255$ for Tiny-ImageNet/ImageNet datasets during adversarial training and when measuring the adversarial test accuracy. Other details for generating adversarial examples and additional experiments on various attacker algorithms are described in Appendix.

**Evaluation metric.** We evaluate the adversarial test accuracy as in [12], which is the classification accuracy on the corrupted test dataset compromised by an attacker (e.g., average attack). We also measure the clean test accuracy using the original clean test data and report the results in Appendix.

**Baselines.** We compare our NEO-KD with the following baselines. First, we consider the scheme based on adversarial training without any knowledge distillation, where adversarial examples are generated by max-average attack or average attack [12]. The second baseline is the conventional self-knowledge distillation (SKD) strategy [20, 24] combined with adversarial training: during adversarial training, the prediction of the last exit for a given clean/adversarial data is distilled to the predictions of all the previous exits for the same clean/adversarial data. The third baseline is the knowledge distillation scheme for adversarial training [8], which distills the prediction of clean data to the prediction of adversarial examples in single-exit networks. As in [8], we distill the last output of clean data to the last output of adversarial data. The last baseline is a regularizer-based adversarial training strategy for multi-exit networks [3], where the regularizer restricts the weights of the fully connected layers (classifiers). In Appendix we compare our NEO-KD with the recent work TEAT [7], a general defense algorithm for single-exit networks.

**Inference scenarios.** At inference time, we consider two widely known setups for multi-exit networks: (i) anytime prediction setup and (ii) budgeted prediction setup. In the anytime prediction setup, an appropriate exit is selected depending on the current latency constraint. In this setup, for each exit, we report the average performance computed with all test samples. In the budgeted prediction setup, given a fixed computational budget, each sample is predicted at different exits depending on the predetermined confidence threshold (which is determined by validation set). Starting from the first exit, given a test sample, when the confidence at the exit (defined as the maximum softmax value) is larger than the threshold, prediction is made at this exit. Otherwise, the sample proceeds to the next exit. In this scenario, easier samples are predicted at earlier exits and harder samples are predicted at later exits, which leads to efficient inference. Given the fixed computation budget and confidence threshold, we measure the average accuracy of the test samples. We evaluate our method in these two

| | top-1 accuracy (%) | | | | | | | | top-5 accuracy (%) | | | | | | | |
|---|---|---|---|---|---|---|---|---|---|---|---|---|---|---|---|---|
| Exit | 1 | 2 | 3 | 4 | 5 | 6 | 7 | Avg. | 1 | 2 | 3 | 4 | 5 | 6 | 7 | Avg. |
| Adv. w/o Distill [12] | 28.04 | 28.32 | 28.34 | 27.64 | 26.78 | 25.93 | 24.77 | 27.12 | **60.64** | **61.15** | 59.13 | 58.46 | 58.38 | 57.46 | 57.44 | 58.95 |
| SKD [24] | 27.36 | 27.68 | 25.79 | 24.74 | 24.10 | 20.86 | 19.28 | 24.26 | 60.50 | 60.23 | 57.56 | 55.98 | 55.11 | 52.71 | 52.17 | 56.32 |
| ARD [8] | 27.60 | 28.00 | 27.99 | 27.42 | 28.32 | 27.25 | 29.08 | 27.95 | 60.26 | 60.51 | 59.31 | 58.40 | 58.40 | 58.09 | **60.62** | 59.37 |
| LW [3] | 20.44 | 20.57 | 19.86 | 19.34 | 19.51 | 19.46 | 20.22 | 19.91 | 50.69 | 50.91 | 48.47 | 48.56 | 48.13 | 49.61 | 49.76 | 49.45 |
| NEO-KD (ours) | **28.37** | **28.78** | **29.02** | **29.49** | **30.06** | **28.45** | **28.54** | **28.96** | 59.58 | 60.04 | **59.67** | **59.29** | **60.46** | **58.96** | 59.44 | **59.63** |

(a) Max-average atack

| | top-1 accuracy (%) | | | | | | | | top-5 accuracy (%) | | | | | | | |
|---|---|---|---|---|---|---|---|---|---|---|---|---|---|---|---|---|
| Exit | 1 | 2 | 3 | 4 | 5 | 6 | 7 | Avg. | 1 | 2 | 3 | 4 | 5 | 6 | 7 | Avg. |
| Adv. w/o Distill [12] | 16.74 | 17.33 | 19.05 | 19.47 | 19.06 | 18.15 | 17.12 | 18.13 | 47.31 | 48.34 | 50.78 | 50.75 | 50.80 | 49.89 | 47.10 | 49.28 |
| SKD [24] | 18.13 | 18.45 | 19.53 | 19.87 | 19.54 | 16.67 | 14.21 | 18.06 | **49.85** | 50.72 | 51.17 | 51.91 | 51.73 | 47.32 | 43.18 | 49.41 |
| ARD [8] | 16.63 | 17.16 | 19.13 | 19.74 | 19.70 | 18.92 | 19.83 | 18.73 | 46.90 | 48.60 | 50.44 | 51.28 | 50.93 | 49.14 | 49.01 | 49.47 |
| LW [3] | 13.78 | 13.95 | 15.16 | 15.62 | 15.86 | 13.01 | 13.57 | 14.42 | 41.14 | 41.27 | 43.29 | 43.67 | 44.08 | 40.49 | 40.66 | 42.09 |
| NEO-KD (ours) | **20.41** | **21.32** | **23.27** | **24.30** | **24.47** | **23.99** | **22.39** | **22.88** | 49.48 | **51.47** | **53.13** | **55.01** | **54.69** | **54.32** | **52.11** | **52.89** |

(b) Average attack

Table 3: **Anytime prediction setup**: Adversarial test accuracy on CIFAR-100.

| | top-1 accuracy (%) | | | | | | top-5 accuracy (%) | | | | | |
|---|---|---|---|---|---|---|---|---|---|---|---|---|
| Exit | 1 | 2 | 3 | 4 | 5 | Avg. | 1 | 2 | 3 | 4 | 5 | Avg. |
| Adv. w/o Distill [12] | 31.30 | 32.38 | 32.52 | 31.42 | 31.56 | 31.84 | 60.20 | 60.84 | 61.10 | 58.64 | 59.56 | 60.07 |
| SKD [24] | **33.04** | 33.24 | 30.50 | 28.40 | 28.50 | 30.74 | **63.68** | 62.72 | 60.04 | 58.50 | 58.28 | 60.64 |
| ARD [8] | 30.08 | 31.84 | 30.56 | 31.22 | 32.34 | 31.21 | 59.88 | 60.74 | 59.62 | 59.28 | 59.72 | 59.85 |
| LW [3] | 26.06 | 26.00 | 24.54 | 24.42 | 24.88 | 25.18 | 53.24 | 51.90 | 51.02 | 51.02 | 51.14 | 51.66 |
| NEO-KD (ours) | 32.96 | **35.08** | **33.42** | **32.40** | **32.64** | **33.30** | 62.48 | **62.82** | **61.76** | **60.70** | **60.92** | **61.74** |

(a) Max-average attack

| | top-1 accuracy (%) | | | | | | top-5 accuracy (%) | | | | | |
|---|---|---|---|---|---|---|---|---|---|---|---|---|
| Exit | 1 | 2 | 3 | 4 | 5 | Avg. | 1 | 2 | 3 | 4 | 5 | Avg. |
| Adv. w/o Distill [12] | 25.22 | 27.34 | 28.94 | 28.06 | 28.48 | 27.61 | 53.68 | 55.46 | 58.18 | 57.46 | 57.38 | 56.43 |
| SKD [24] | **28.40** | 29.26 | 28.74 | 28.14 | 26.82 | 28.27 | **58.34** | 59.30 | 58.38 | 57.72 | 55.48 | 57.84 |
| ARD [8] | 24.48 | 26.76 | 27.78 | 28.46 | 29.14 | 27.32 | 53.44 | 56.10 | 57.20 | 57.32 | 57.08 | 56.23 |
| LW [3] | 22.34 | 23.12 | 23.58 | 22.76 | 23.30 | 23.02 | 47.56 | 48.16 | 49.86 | 48.64 | 49.72 | 48.79 |
| NEO-KD (ours) | 28.24 | **31.14** | **30.58** | **31.58** | **31.24** | **30.56** | 57.34 | **59.82** | **60.16** | **60.04** | **59.06** | **59.28** |

(b) Average attack

Table 4: **Anytime prediction setup**: Adversarial test accuracy on Tiny-ImageNet.

| | top-1 accuracy (%) | | | | | | top-5 accuracy (%) | | | | | |
|---|---|---|---|---|---|---|---|---|---|---|---|---|
| Exit | 1 | 2 | 3 | 4 | 5 | Avg. | 1 | 2 | 3 | 4 | 5 | Avg. |
| Adv. w/o Distill [12] | 26.10 | 31.71 | 31.94 | 30.22 | 32.30 | 30.45 | 55.40 | 59.84 | 59.23 | 57.17 | 59.90 | 58.31 |
| SKD [24] | 27.34 | 31.44 | 29.63 | 27.03 | 26.71 | 28.43 | **56.96** | **59.94** | 57.49 | 53.98 | 55.24 | 56.72 |
| ARD [8] | 25.81 | 32.00 | 32.93 | 32.00 | 31.72 | 30.89 | 54.46 | 59.34 | 59.24 | 57.93 | 58.29 | 57.85 |
| NEO-KD (ours) | **27.89** | **32.61** | **32.99** | **32.74** | **35.63** | **32.37** | 55.46 | 59.64 | **59.79** | **59.61** | **62.46** | **59.39** |

(a) Max-average attack

| | top-1 accuracy (%) | | | | | | top-5 accuracy (%) | | | | | |
|---|---|---|---|---|---|---|---|---|---|---|---|---|
| Exit | 1 | 2 | 3 | 4 | 5 | Avg. | 1 | 2 | 3 | 4 | 5 | Avg. |
| Adv. w/o Distill [12] | 18.20 | 24.21 | 28.24 | 28.73 | 28.50 | 25.58 | 44.74 | 52.57 | 56.82 | 57.55 | 57.14 | 53.77 |
| SKD [24] | 19.04 | 24.45 | 26.45 | 25.72 | 22.17 | 23.57 | 46.31 | 52.92 | 55.33 | 54.71 | 50.18 | 51.89 |
| ARD [8] | 17.56 | 23.68 | 27.98 | 28.52 | 25.83 | 24.71 | 43.76 | 52.07 | 56.42 | 57.30 | 53.48 | 52.60 |
| NEO-KD (ours) | **22.42** | **28.62** | **31.77** | **32.78** | **34.30** | **29.98** | **48.02** | **55.30** | **58.76** | **60.07** | **61.35** | **56.70** |

(b) Average attack

Table 5: **Anytime prediction setup**: Adversarial test accuracy on ImageNet.

settings and show that our method outperforms the baselines in both settings. More detailed settings for our inference scenarios are provided in Appendix.

## 4.2 Main Experimental Results

**Result 1: Anytime prediction setup.** Tables 1, 2, 3, 4, 5 compare the adversarial test accuracy of different schemes under max-average attack and average attack using MNIST, CIFAR-10/100, Tiny-ImageNet, and ImageNet, respectively. Note that we achieve adversarial accuracies between 40% - 50% for CIFAR-10, which is standard considering the prior works on robust multi-exit networks [12]. Our first observation from the results indicates that the performance of SKD [24] is generally lower than that of *Adv. w/o Distill*, whereas ARD [8] outperforms *Adv. w/o Distill*. This suggests that the naive application of self-knowledge distillation can either increase or decrease the adversarial robustness of multi-exit networks. Consequently, the method of knowledge distillation

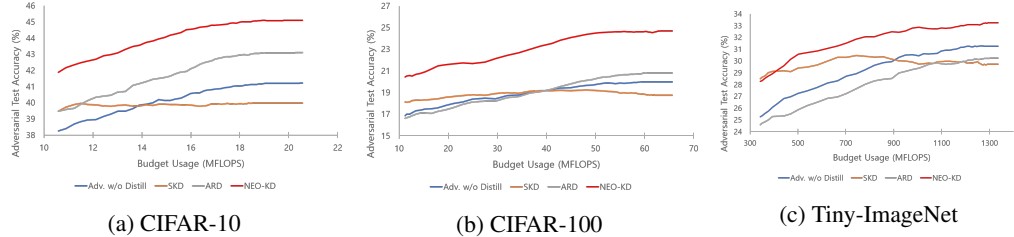

(a) CIFAR-10        (b) CIFAR-100        (c) Tiny-ImageNet

Figure 2: **Budgeted prediction setup**: Adversarial test accuracy under average attack. The result for LW is excluded since the performance is too low and thus hinders the comparison between baselines and NEO-KD.

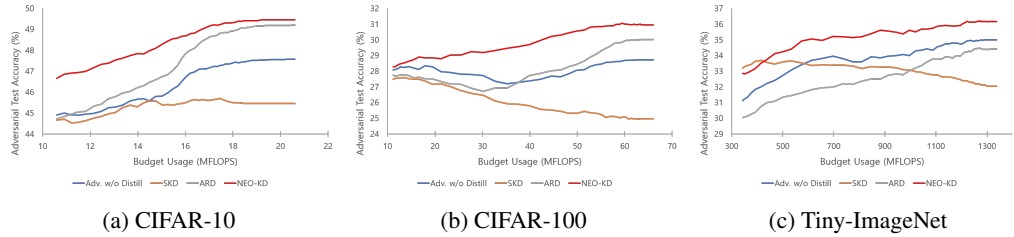

(a) CIFAR-10        (b) CIFAR-100        (c) Tiny-ImageNet

Figure 3: **Budgeted prediction setup**: Adversarial test accuracy under max-average attack. The result for LW is excluded since the performance is too low and thus hinders the comparison between baselines and NEO-KD.

significantly influences the robustness of multi-exit networks (i.e., determining which knowledge to distill and which exit to target). To further enhance robustness, we investigate strategies for distilling high-quality knowledge and mitigating adversarial transferability.

By combining EOKD with NKD to mitigate dependency across submodels while guiding a multi-exit network to extract high quality features from adversarial examples as original data, NEO-KD achieves the highest adversarial test accuracy at most exits compared to the baselines for all datasets/attacks. The overall results confirm the advantage of NEO-KD for robust multi-exit networks.

**Result 2: Budgeted prediction setup.** Different from the anytime prediction setup where the pure performance of each exit is measured, in this setup, we adopt *ensemble strategy* at inference time where the predictions from the selected exit (according to the confidence threshold) and the previous exits are ensembled. From the results in anytime prediction setup, it is observed that various schemes tend to show low performance at the later exits compared to earlier exits in the model, where more details are discussed in Appendix. Therefore, this *ensemble strategy* can boost the performance of the later exits. With the ensemble scheme, given a fixed computation budget, we compare adversarial test accuracies of our method with the baselines.

Figures 2 and 3 show the results in budgeted prediction setup under average attack and max-average attack, respectively. NEO-KD achieves the best adversarial test accuracy against both average and max-average attacks in all budget setups. Our scheme also achieves the target accuracy with significantly smaller computing budget compared to the baselines. For example, to achieve $41.21\%$ of accuracy against average attack using CIFAR-10 (which is the maximum accuracy of *Adv. w/o Distill*), the proposed NEO-KD needs $10.59$ MFlops compared to Adv. w/o Distill that requires $20.46$ MFlops, saving $48.24\%$ of computing budget. Compared to ARD, our NEO-KD saves $25.27\%$ of computation, while SKD and LW are unable to achieve this target accuracy. For CIFAR-100 and Tiny-ImageNet, NEO-KD saves $81.60\%$ and $50.27\%$ of computing budgets compared to *Adv. w/o Distill*. The overall results are consistent with the results in anytime prediction setup, confirming the advantage of our solution in practical settings with limited computing budget.

**Result 3: Adversarial transferability.** We also compare the adversarial transferability of our NEO-KD and different baselines among exits in a multi-exit neural network. When measuring adversarial transferability, as in [29], we initially gather all clean test samples for which all exits produce correct predictions. Subsequently, we generate adversarial examples targeting each exit using the collected clean samples (We use PGD-50 based single attack). Finally, we assess the adversarial transferability as the attack success rate of these adversarial examples at each exit. Figure 4 shows the adversarial transferability map of each scheme on CIFAR-100. Here, each row corresponds to the target exit for generating adversarial examples, and each column corresponds the exit where attack success rate is measured. For example, the $(i, j)$-th element in the map is adversarial transferability measured at exit

| | Exit 1 | Exit 2 | Exit 3 | Exit 4 | Exit 5 | Exit 6 | Exit 7 |
|---|---|---|---|---|---|---|---|
| Exit 1 | 81.55 | 29.40 | 21.23 | 17.77 | 14.96 | 12.59 | 12.89 |
| Exit 2 | 33.73 | 79.43 | 25.10 | 21.53 | 18.28 | 15.15 | 14.88 |
| Exit 3 | 30.33 | 31.04 | 73.87 | 30.52 | 26.29 | 21.01 | 21.36 |
| Exit 4 | 27.30 | 27.52 | 32.40 | 73.32 | 30.35 | 24.63 | 23.38 |
| Exit 5 | 23.27 | 23.43 | 26.81 | 29.48 | 75.40 | 28.26 | 28.15 |
| Exit 6 | 19.86 | 18.37 | 19.81 | 22.53 | 27.71 | 76.84 | 41.66 |
| Exit 7 | 16.38 | 15.01 | 15.89 | 17.63 | 21.01 | 35.59 | 80.03 |

(a) Adv. w/o Distill [12] (Avg. w/o Diag.: 23.68%)

| | Exit 1 | Exit 2 | Exit 3 | Exit 4 | Exit 5 | Exit 6 | Exit 7 |
|---|---|---|---|---|---|---|---|
| Exit 1 | 79.38 | 38.22 | 29.08 | 24.39 | 20.94 | 19.78 | 19.52 |
| Exit 2 | 43.55 | 76.42 | 36.37 | 30.56 | 26.36 | 24.11 | 24.15 |
| Exit 3 | 38.49 | 41.15 | 71.50 | 41.78 | 36.73 | 32.08 | 31.40 |
| Exit 4 | 37.24 | 38.46 | 44.49 | 69.32 | 44.56 | 37.77 | 36.85 |
| Exit 5 | 33.21 | 34.21 | 38.87 | 43.72 | 68.91 | 43.36 | 41.63 |
| Exit 6 | 24.51 | 25.64 | 28.16 | 31.84 | 36.40 | 75.41 | 63.31 |
| Exit 7 | 18.92 | 18.49 | 20.36 | 22.50 | 25.35 | 52.70 | 81.18 |

(b) SKD [24] (Avg. w/o Diag.: 33.36%)

| | Exit 1 | Exit 2 | Exit 3 | Exit 4 | Exit 5 | Exit 6 | Exit 7 |
|---|---|---|---|---|---|---|---|
| Exit 1 | 71.88 | 33.64 | 22.36 | 16.92 | 12.78 | 10.34 | 10.83 |
| Exit 2 | 38.66 | 66.96 | 28.06 | 21.09 | 16.07 | 12.80 | 12.72 |
| Exit 3 | 30.63 | 31.77 | 63.57 | 29.57 | 21.72 | 17.26 | 16.77 |
| Exit 4 | 28.14 | 27.83 | 34.28 | 58.28 | 28.38 | 21.77 | 21.33 |
| Exit 5 | 23.27 | 23.58 | 25.91 | 28.35 | 60.77 | 28.27 | 25.03 |
| Exit 6 | 18.58 | 18.24 | 19.36 | 20.96 | 26.04 | 63.88 | 40.48 |
| Exit 7 | 14.72 | 13.94 | 14.82 | 15.76 | 19.20 | 33.56 | 70.12 |

(c) NKD (Avg. w/o Diag.: 22.76%)

| | Exit 1 | Exit 2 | Exit 3 | Exit 4 | Exit 5 | Exit 6 | Exit 7 |
|---|---|---|---|---|---|---|---|
| Exit 1 | 66.68 | 31.06 | 20.19 | 15.37 | 11.61 | 9.41 | 10.12 |
| Exit 2 | 35.34 | 62.46 | 24.30 | 17.85 | 13.82 | 11.61 | 12.41 |
| Exit 3 | 26.95 | 27.45 | 58.90 | 25.24 | 18.46 | 15.14 | 15.48 |
| Exit 4 | 25.66 | 24.44 | 28.11 | 54.76 | 24.58 | 19.72 | 19.48 |
| Exit 5 | 21.02 | 19.83 | 21.27 | 24.14 | 57.13 | 23.37 | 22.32 |
| Exit 6 | 17.66 | 16.83 | 17.27 | 18.98 | 23.01 | 59.20 | 35.67 |
| Exit 7 | 13.57 | 12.83 | 12.80 | 13.63 | 16.61 | 30.43 | 66.37 |

(d) NEO-KD (ours) (Avg. w/o Diag.: 20.12%)

Figure 4: **Adversarial transferability map** of each method on CIFAR-100. Diag. indicates the diagonal of the matrix. **Row**: Target exit for generating adversarial examples. **Column**: Exit where adversarial transferability is measured. Adopting NKD solely already achieves better adversarial transferability compared to the existing baselines. Applying EOKD to NKD can further improve adversarial transferability by reducing the dependency among different submodels in the multi-exit network.

| Exit | 1 | 2 | 3 | Average |
|---|---|---|---|---|
| Adv. w/o Distill | 38.00% | 41.42% | 40.70% | 40.04% |
| NKD | 40.01% | 42.67% | 40.23% | 40.97% |
| EOKD | 37.26% | 41.10% | 38.68% | 39.01% |
| NEO-KD (ours) | **41.67%** | **45.38%** | **45.54%** | **44.20%** |

Table 6: **Effect of each component of NEO-KD**: Adversarial test accuracy against average attack on CIFAR-10. NKD and EOKD work in a complementary fashion and have implicit synergies.

| Type of ensembles | 1 | 2 | 3 | Average |
|---|---|---|---|---|
| Adv. w/o Distill | 38.00% | 41.42% | 40.70% | 40.04% |
| No ensembling | 39.54% | 41.77% | 40.15% | 40.49% |
| Ensemble neighbors (NKD) | **41.67%** | **45.38%** | **45.54%** | **44.20%** |
| Ensemble all exits | 38.63% | 41.61% | 39.93% | 40.06% |

Table 7: **Results with different number of ensembles**: Adversarial test accuracy against average attack on CIFAR-10. The neighbor-wise ensemble case corresponds to our NKD.

$j$, generated by the adversarial examples targeting exit $i$. The values and the brightness in the map indicate success rates of attacks; lower value (dark color) means lower adversarial transferability.

We have the following key observations from adversarial transferability map. First, as observed in Figure 4a, compared to *Adv. w/o Distill* [12], SKD [24] in Figure 4b exhibits higher adversarial transferability. This indicates that distilling the same teacher prediction to every exit leads to a high dependency across exits. Thus, it is essential to consider distilling non-overlapping knowledge across different exits. Second, when compared to the baselines [12, 24], our proposed NKD in Figure 4c demonstrates low adversarial transferability. This can be attributed to the fact that NKD takes into account the quality of the distilled features and ensures that the features are not overlapping among exits. Third, as seen in Figure 4d, the adversarial transferability is further mitigated by incorporating EOKD, which distills orthogonal class predictions to different exits, into NKD. Comparing the average of attack success rate across all exits (excluding the values of the target exits shown in the diagonal), it becomes evident that NEO-KD yields $3.56\%$ and $13.24\%$ gains compared to *Adv. w/o Distill* and SKD, respectively. The overall results confirm the advantage of our solution to reduce the adversarial transferability in multi-exit networks. These results support the improved adversarial test accuracy of NEO-KD reported in Section 4.2.

### 4.3 Ablation Studies and Discussions

**Effect of each component of NEO-KD.** In Table 6, we observe the effects of our individual components, NKD and EOKD. It shows that combining NKD and EOKD boosts up the performance beyond the sum of their original gains. Given different roles, the combination of NKD and EOKD enables multi-exit networks to achieve the state-of-the-art performance under adversarial attacks.

**Effect of the type of ensembles in NKD.** In the proposed NKD, we consider only the neighbor exits to distill the knowledge of clean data. What if we consider fewer or more exits than neighboring exits? If the number of ensembles is too small, the scheme does not distill high-quality features. If the number of ensembles is too large, the dependencies among submodels increase, resulting in high adversarial transferability. To see this effect, in Table 7, we measure adversarial test accuracy of three types of ensembling methods depending on the number of exits used for constructing ensembles: *no ensembling*, *ensemble neighbors (NKD)*, and *ensemble all exits*. In *no enesmbling* approach, we distill the knowledge of each exit from clean data to the output at the same position of exit for adversarial examples. In contrast, the *ensemble all exits* scheme averages the knowledge of all exits from clean data and provides it to all exits of adversarial examples. The *ensemble neighbors* approach corresponds to our NKD. The results show that the proposed NEO-KD with neighbor ensembling

| Attacks | Exit | 1 | 2 | 3 | 4 | 5 | 6 | 7 | Average |
|---------|------|---|---|---|---|---|---|---|---------|
| PGD-100 | Adv. w/o Distill [12] | 16.82% | 17.24% | 19.03% | 19.41% | 18.97% | 17.90% | 16.96% | 18.05% |
|  | NEO-KD (ours) | **20.38%** | **21.29%** | **23.22%** | **24.38%** | **24.44%** | **23.82%** | **22.21%** | **22.82%** |
| CW | Adv. w/o Distill [12] | **35.31%** | **35.72%** | 36.47% | 36.85% | 37.20% | 37.19% | 36.68% | 36.49% |
|  | NEO-KD (ours) | 31.56% | 34.55% | **38.20%** | **40.60%** | **43.48%** | **44.03%** | **43.01%** | **39.35%** |
| AutoAttack | Adv. w/o Distill [12] | 31.32% | 34.24% | 37.27% | 39.67% | 41.20% | 41.73% | 40.45% | 37.98% |
|  | NEO-KD (ours) | **31.56%** | **34.55%** | **38.20%** | **40.60%** | **43.48%** | **44.03%** | **43.01%** | **39.35%** |

Table 8: **Results with stronger attacker algorithms**: Adversarial test accuracy against average attack using CIFAR-100 dataset.

enables to distill high-quality features while lowering dependencies among submodels, confirming our intuition.

**Robustness against stronger adversarial attack.** We evaluate NEO-KD against stronger adversarial attacks; we perform average attack based on PGD-100 [21], Carlini and Wagner (CW) [2], and AutoAttack [5]. Table 8 shows that NEO-KD achieves higher adversarial test accuracy than *Adv. w/o Distill* [12] in most of cases. Typically, CW attack and AutoAttack are stronger attacks than the PGD attack in single-exit networks. However, in the context of multi-exit networks, these attacks become weaker than the PGD attack when taking all exits into account. Details for generating stronger adversarial attacks are described in Appendix.

**Additional results.** Other results including clean test accuracy, results with average attack based adversarial training, results with varying hyperparameters, and results with another baseline used in single-exit network, are provided in Appendix.

## 5 Conclusion

In this paper, we proposed a new knowledge distillation based adversarial training strategy for robust multi-exit networks. Our solution, NEO-KD, reduces adversarial transferability in the network while guiding the output of the adversarial examples to closely follow the ensemble outputs of the neighbor exits of the clean data, significantly improving the overall adversarial test accuracy. Extensive experimental results on both anytime and budgeted prediction setups using various datasets confirmed the effectiveness of our method, compared to baselines relying on existing adversarial training or knowledge distillation techniques for multi-exit networks.

## Acknowledgement

This work was supported by the National Research Foundation of Korea (NRF) grant funded by the Korea government (MSIT) (No. NRF-2019R1I1A2A02061135), by the Center for Applied Research in Artificial Intelligence (CARAI) grant funded by DAPA and ADD (UD230017TD), and by IITP funds from MSIT of Korea (No. 2020-0-00626).

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

# A  Experiment Details

We provide additional implementation details that were not described in the main manuscript.

## A.1  Model training

We train a SmallCNN [12] for 150 epochs with batch size 128 on MNIST [18], a MSDNet [13] for 150 epochs with batch size 128 on CIFAR-10/100 [16] and Tiny-ImageNet [17]. For ImageNet [25], since it takes a long time to train from the beginning, we finetune the pretrained model with our NEO-KD loss function for 10 epochs. All experiments are implemented with two RTX3090 GPUs. Other settings follow [12][2] for SmallCNN and [13][3] for MSDNet except for a channel, which is set to 8 for our experiments on CIFAR-10/100. For the optimizer, SGD is used with a momentum of 0.9 and a weight decay of $5 \times 10^{-4}$. For the MNIST dataset, the initial learning rate is set to 0.01 and the learning rate is decayed 10-fold at 50 epoch. For the CIFAR-10/100 dataset, the initial learning rate is 0.1, and is decayed 10-fold at 75-th epoch and 115-th epoch. For Tiny-ImageNet, the initial learning rate is set to 0.1, and is decayed 10-fold at 50-th epoch and 100-th epoch. For ImageNet, the learning rate is constant with 0.001.

## A.2  Adversarial training

During adversarial training, we use max-average attack and average attack [12] for generating adversarial examples via PGD attacker algorithm [21] with 7-steps, while the PGD attacker algorithm with 50-step is used for measuring robustness against a stronger attack during test time. With PGD-50 attack, we measure adversarial test accuracy on 3 random seeds and average the results. For PGD attack, the perturbation degree $\epsilon = 0.3$ is used for MNIST, $\epsilon = 8/255$ is used for CIFAR-10/100, and $\epsilon = 2/255$ Tiny-ImageNet/ImageNet during both adversarial training and inference time. The step size $\delta$ is set to $20/255$ for MNIST, $2/255$ for CIFAR-10/100, and $\frac{2}{3}\epsilon$ (0.0052) for Tiny-ImageNet/ImageNet. The number of iterations is commonly 50-steps. Similarly, when measuring adversarial transferability, we also use PGD attack with 50 steps, $2/255$ step size, and $8/255$ epsilon value to generate adversarial attacks. With PGD attack, the attack success rate is utilized as the metric to measure adversarial transferability. We perform each experiment on 5 random seeds and average the results.

Additionally, the hyperparameter $\alpha$ for NKD is set to 3, and $\beta$ for EOKD is set to 1 across all experiments. On the other hand, the exit-balancing parameter $\gamma$ is set to $[1, 1, 1]$ for MNIST and CIFAR-10, and $[1, 1, 1, 1.5, 1.5]$, $[1, 1, 1, 1.5, 1.5, 1.5, 1.5]$ for Tiny-ImageNet/ImageNet, CIFAR-100, respectively.

## A.3  How to determine confidence threshold in budgeted prediction setup

We provide a detailed explanation about how to determine confidence threshold for each exit using validation set before the testing phase. First, in order to obtain confidence thresholds for various budget scenarios, we allocate the number of validation samples for each exit. For simplicity, consider a toy example with 3-exit network (i.e., $L = 3$) and assume the number of validation set is 3000. Then, each exit can be assigned a different number of samples: for instance, (2000, 500, 500), (1000, 1000, 1000) and (500, 1000, 1500). As more samples are allocated to the early exits, a scenario with a smaller budget can be obtained, while allocating more data to the later exits can lead to a scenario with a larger budget. More specifically, to see how to obtain the confidence threshold for each exit, consider the low-budget case of (2000, 500, 500). The model first makes predictions on all 3000 samples at exit 1 and sorts the samples based on their confidence. Then, the 2000-th largest confidence value is set as the confidence threshold for the exit 1. Likewise, the model performs predictions on remaining 1000 samples at exit 2 and the 500-th largest confidence is determined as the threshold for exit 2. Following this process, all thresholds for each exit are determined. During the testing phase, we perform predictions on test samples based on the predefined thresholds for each exit, and calculate the total computational budget for the combination of (2000, 500, 500). In this way, we can obtain accuracy and computational budget for different combinations of data numbers

---

[2]https://github.com/VITA-Group/triple-wins
[3]https://github.com/kalviny/MSDNet-PyTorch

| Exit | 1 | 2 | 3 | Average |
|------|-----|-----|-----|---------|
| Adv. w/o Distill [12] | 98.14% | 99.36% | 99.55% | 99.02% |
| NEO-KD (ours) | 97.82% | 99.29% | 99.48% | 98.86% |

(a) MNIST

| | top-1 accuracy (%) | | | | | | top-5 accuracy (%) | | | | | |
|------|-----|-----|-----|-----|-----|------|-----|-----|-----|-----|-----|------|
| Exit | 1 | 2 | 3 | 4 | 5 | Avg. | 1 | 2 | 3 | 4 | 5 | Avg. |
| Adv. w/o Distill [12] | 43.10 | 47.64 | 50.36 | 50.58 | 50.70 | 48.48 | 68.26 | 72.30 | 74.86 | 75.58 | 75.36 | 73.27 |
| NEO-KD (ours) | 45.56 | 48.32 | 49.88 | 50.56 | 50.52 | 48.97 | 70.56 | 73.12 | 74.52 | 74.22 | 74.98 | 73.48 |

(b) Tiny-ImageNet

Table A1: **Clean test accuracy in the anytime prediction setup:** NEO-KD's advantage in terms of adversarial test accuracy (as shown in the main mansuscript) can be achieved without largely compromising the clean test accuracy.

| | top-1 accuracy (%) | | | | | | | | top-5 accuracy (%) | | | | | | | |
|------|-----|-----|-----|-----|-----|-----|-----|------|-----|-----|-----|-----|-----|-----|-----|------|
| Exit | 1 | 2 | 3 | 4 | 5 | 6 | 7 | Avg. | 1 | 2 | 3 | 4 | 5 | 6 | 7 | Avg. |
| Adv. w/o Distill [12] | 21.14 | 20.49 | 20.68 | 20.18 | 20.58 | 20.56 | 20.24 | 20.55 | 48.28 | 48.24 | 46.75 | 46.41 | 46.31 | 46.96 | 45.64 | 46.94 |
| SKD [24] | **22.39** | **22.17** | 19.89 | 18.81 | 18.82 | 17.48 | 14.84 | 19.20 | **51.73** | **51.43** | 47.88 | 45.92 | 45.90 | 43.65 | 41.38 | 46.84 |
| ARD [8] | 20.55 | 19.84 | 20.95 | 19.61 | 19.41 | 20.52 | **21.72** | 20.37 | 48.18 | 46.94 | 46.96 | 45.37 | 45.06 | 46.17 | **48.42** | 46.73 |
| LW [3] | 17.65 | 18.80 | 16.97 | 16.64 | 17.15 | 17.08 | 16.79 | 17.30 | 44.33 | 44.03 | 41.49 | 41.48 | 41.37 | 42.06 | 41.64 | 42.34 |
| NEO-KD (ours) | 21.31 | 22.01 | **21.46** | **22.10** | **21.89** | **21.64** | 20.78 | **21.60** | 49.16 | 48.88 | **49.12** | **48.76** | **48.00** | **48.08** | 47.23 | **48.46** |

(a) Max-average attack

| | top-1 accuracy (%) | | | | | | | | top-5 accuracy (%) | | | | | | | |
|------|-----|-----|-----|-----|-----|-----|-----|------|-----|-----|-----|-----|-----|-----|-----|------|
| Exit | 1 | 2 | 3 | 4 | 5 | 6 | 7 | Avg. | 1 | 2 | 3 | 4 | 5 | 6 | 7 | Avg. |
| Adv. w/o Distill [12] | 12.73 | 12.10 | 13.33 | 13.27 | 13.14 | 13.17 | 13.27 | 13.00 | 36.22 | 35.66 | 37.07 | 36.57 | 36.47 | 37.26 | 36.51 | 36.54 |
| SKD [24] | **15.96** | **15.90** | 15.31 | 15.32 | 15.70 | 14.84 | 10.33 | 14.77 | **42.19** | **41.72** | 41.29 | 41.46 | **42.03** | 40.08 | 32.68 | **40.21** |
| ARD [8] | 12.74 | 12.53 | 13.80 | 13.04 | 12.79 | 13.56 | 14.73 | 13.31 | 36.12 | 35.67 | 37.21 | 36.28 | 35.81 | 37.09 | 38.82 | 36.71 |
| LW [3] | 12.82 | 13.33 | 12.28 | 11.96 | 12.10 | 12.25 | 12.55 | 12.47 | 36.49 | 36.23 | 34.46 | 34.17 | 34.09 | 34.78 | 34.72 | 34.99 |
| NEO-KD (ours) | 14.64 | 15.15 | **15.42** | **16.07** | **16.42** | **16.11** | **15.59** | **15.63** | 38.88 | 39.56 | 40.60 | 40.57 | 41.13 | **40.91** | **39.85** | **40.21** |

(b) Average attack

Table A2: **Adversarial training via Average attack:** Adversarial test accuracy in the anytime prediction setup on CIFAR-100.

(i.e., various budget scenarios). Figures 2 and 3 in the main manuscript show the results for 100 cases of different budget scenarios.

# B  Clean Test Accuracy

Table A1 shows the clean accuracy results using the model built upon adversarial training via max-average attack. We observe that NEO-KD generally shows comparable clean test accuracy with *Adv. w/o Distill* [12], especially on the more complicated dataset Tiny-ImageNet [17] while achieving much better adversarial test accuracy as reported in the main manuscript.

# C  Adversarial Training via Average Attack

In the main manuscript, we presented experimental results using the model trained based on max-average attack. Here, we also adversarially train the model via average attack [12] and measure adversarial test accuracy on CIFAR-100 dataset. Table A2 compares adversarial test accuracies of NEO-KD and other baselines against max-average attack and average attack. The overall results are consistent with the ones in the main manuscript with adversarial training via max-average attack, further confirming the advantage of NEO-KD.

# D  Hyperparameter Tuning

In the NEO-KD objective function, there are three hyperparameters $(\alpha, \beta, \gamma)$, where $\alpha$, $\beta$ control the amount of distilling knowledge from NKD, EOKD and $\gamma$ increases the amount of knowledge distilled to later exits.

**D.1 Hyperparameter** $(\alpha, \beta)$

The extreme value of $\alpha$ and $\beta$ can destroy ideal adversarial training. Too large $\alpha$ makes strong NKD, which results in high dependency among submodels and too small $\alpha$ makes weak NKD, which cannot distill enough knowledge to student exits. In contrast, too large $\beta$ makes strong EOKD, which can interrupt adversarial training by distilling only sparse knowledge (likelihoods of majority classes are zero) and too small $\beta$ makes weak EOKD, which cannot mitigate dependency among submodels. We select $\alpha$, $\beta$ values in the range of $[0.35, 3]$ and measure the adversarial test accuracy value by averaging adversarial test accuracy from all exits. The candidate $(\alpha, \beta)$ pairs are $(0.35, 1)$, $(1, 0.35)$, $(0.35, 0.35)$, $(0.5, 1)$, $(1, 0.5)$, $(0.5, 0.5)$, $(1, 1)$, $(2, 1)$, $(1, 2)$, $(2, 2)$, $(3, 1)$, $(1, 3)$, and $(3, 3)$. When $(\alpha, \beta)$ is $(3, 1)$, NEO-KD achieves 28.96% of adversarial test accuracy against max-average attack and 22.88% against average attack, which is the highest adversarial test accuracy among the various candidate $(\alpha, \beta)$ pairs. Therefore, we use $(3, 1)$ as $(\alpha, \beta)$ pair in our experiments.

**D.2 Hyperparameter** $\gamma$

Since the prediction difference between the last exit (teacher prediction) and later exits is smaller than the prediction difference between the last exit and early exits, later exits are less effective for taking advantage of knowledge distillation. Therefore, we provide slightly larger weights to later exits for distilling more knowledge to later exits than early exits. The candidate $\gamma$ values are $[1, 1, 1, 1, 1, 1, 1]$, $[1, 1, 1, 1.5, 1.5, 1.5, 1.5]$, and $[1, 1, 1, 1.7, 1.7, 1.7, 1.7]$. As a result, when we distill 1.5 times more knowledge to later exits, NEO-KD achieves 28.96% of adversarial test accuracy against max-average attack and 22.88% against average attack, which is the highest adversarial test accuracy compared to providing same weights with earlier exits to later exits (28.13% for max-average and 21.66% for average attack) or distilling 1.7 times more knowledge to later exits than earlier exits (28.68% for max-average and 22.58% for average attack). The adversarial test accuracy value is the average of adversarial test accuracies from all exits. Therefore, we use $\gamma = [1, 1, 1, 1.5, 1.5, 1.5, 1.5]$ in our experiments. This result proves that the exit-balancing parameter $\gamma$ with an appropriate value is needed for high performance.

# E    Discussions on Performance Degradation at Later Exits

As can be seen from the results for the anytime prediction in the main manuscript, the adversarial test accuracy of the later exits is sometimes lower than the performance of earlier exits. This phenomenon can be explained as follows: In general, we observed via experiments that adversarial examples targeting later exits has the higher sum of losses from all exits compared to adversarial examples targeting earlier exits. This makes max-average or average attack mainly focus on attacking the later exits, leading to low adversarial test accuracy at later exits. The performance of later exits can be improved by adopting the ensemble strategy as in the main manuscript for the budgeted prediction setup.

# F    Comparison with Recent Defense Methods for Single-Exit Networks

The baselines in the main paper were generally the adversarial defense methods designed for multi-exit networks. In this section, we conduct additional experiments with a recent defense method, TEAT [7], and compare with our method. Since TEAT was originally designed for the single-exit network, we first adapt TEAT to the multi-exit network setting. Instead of the original TEAT that generates the adversarial examples considering the final output of the network, we modify TEAT to generate adversarial examples that maximizes the average loss of all exits in the multi-exit network. Table A3 below shows the results using max-average attack on CIFAR-10/100. It can be seen that our NEO-KD, which is designed for multi-exit networks, achieves higher adversarial test accuracy compared to the TEAT methods (PGD-TE and TRADES-TE) designed for single-exit networks. The results highlight the necessity of developing adversarial defense techniques geared to multi-exit networks rather than adapting general defense methods used for single-exit networks.

| Exit | 1 | 2 | 3 | Average |
|---|---|---|---|---|
| PGD-TE [7] | **48.73%** | 46.00% | 46.85% | 47.19% |
| TRADES-TE [7] | 45.05% | 39.64% | 42.10% | 42.26% |
| NEO-KD (ours) | 46.53% | **47.65%** | **50.71%** | **48.30%** |

(a) CIFAR-10

| Exit | 1 | 2 | 3 | 4 | 5 | 6 | 7 | Avg. |
|---|---|---|---|---|---|---|---|---|
| PGD-TE [7] | 24.07% | 24.39% | 25.14% | 25.35% | 26.29% | 25.57% | 24.60% | 25.06% |
| TRADE-TE [7] | 17.62% | 18.52% | 18.61% | 18.98% | 18.95% | 19.67% | 20.35% | 18.96% |
| NEO-KD (ours) | **28.37%** | **28.78%** | **29.02%** | **29.49%** | **30.06%** | **28.45%** | **28.54%** | **28.96%** |

(b) CIFAR-100

Table A3: Comparison of adversarial test accuracy against max-average attack between TEAT methods and our NEO-KD.

| Exit | 1 | 2 | 3 | Avg. |
|---|---|---|---|---|
| SKD (exit 1) | 32.27% | 36.92% | 38.57% | 35.92% |
| SKD (exit 2) | 35.33% | 35.10% | 37.82% | 36.08% |
| SKD (exit 3) | 39.36% | 41.39% | 38.39% | 39.71% |
| SKD (ensemble) | 38.63% | 41.80% | 40.13% | 40.19% |
| ARD (exit 1) | 35.64% | 38.10% | 42.12% | 38.62% |
| ARD (exit 2) | 35.35% | 38.24% | 40.00% | 37.86% |
| ARD (exit 3) | 39.37% | 41.98% | 43.53% | 41.63% |
| ARD (ensemble) | 35.22% | 38.35% | 40.76% | 38.11% |
| NEO-KD (ours) | 41.67% | 45.38% | 45.54% | 44.20% |

Table A4: Adversarial test accuracy of SKD and ARD according to exit selection as a teacher prediction.

# G    Comparison with SKD and ARD

Existing self-distillation schemes [20, 24] for multi-exit networks improve the performance on clean samples by self-distilling the knowledge of the last exit, as the last exit has the best prediction quality. Therefore, following the original philosophy, we also used the last exit in implementing the SKD baseline. Regarding ARD [8], since it was proposed for single-exit networks, we also utilized the last exit with high performance when applying ARD to multi-exit networks. Nevertheless, we perform additional experiments to consider comprehensive baselines using various exits for distillation. Table A4 above shows the results of SKD and ARD using a specific exit or an ensemble of all exits for distillation. The results show that our scheme consistently outperforms all baselines.

# H    Implementations of Stronger Attacker Algorithms

In Section 4.3 of the main manuscript, during inference, we replaced the Projected Gradient Descent (PGD) attack with other attacker algorithms (PGD-100 attack, Carlini and Wagner (CW) attack [2], and AutoAttack [5]) to generate stronger attacks for multi-exit neural networks. This section provides explanation on how these stronger attacks are implemented tailored to multi-exit neural networks.

## H.1    Carlini and Wagner (CW) attack

The Carlini and Wagner (CW) attack is a method of generating adversarial examples designed to reduce the difference between the logits of the correct label and the largest logits among incorrect labels. In alignment with this attack strategy, we modify the CW attack for multi-exit neural networks. In the process of minimizing this difference, our modification aims to minimize the average difference across all exits of the multi-exit neural network. Moreover, when deciding whether a sample has been successfully converted into an adversarial example, we consider a sample adversarial if it misleads all exits in the multi-exit neural network.

## H.2 AutoAttack

AutoAttack produces adversarial attacks by ensembling various attacker algorithms. For our experiment, we sequentially use APGD [5], APGD-T [5], FAB [4], and Square [1] algorithms to generate adversarial attacks, as they are commonly used.

### H.2.1 APGD and APGD-T

The APGD attack is a modified version of the PGD attack, which is limited by its fixed step size, a suboptimal choice. The APGD attack overcomes this limitation by introducing an adaptive step size and a momentum term. Similarly, the APGD-T attack is a variation of the APGD attack where the attack perturbs a sample to change to a specific class. In this process, we use the average loss of all exits in the multi-exit neural network as the loss for computing the gradient for adversarial example updates. Moreover, we define a sample as adversarial if it misleads all exits in the multi-exit neural network.

### H.2.2 FAB

The FAB attack creates adversarial attacks through a process involving linear approximation of classifiers, projection to the classifier hyperplane, convex combinations, and extrapolation. The FAB attack first defines a hyperplane classifier separating two classes, then finds a new adversarial example through a convex combination of the extrapolation-projected current adversarial example and the extrapolation-projected original sample with the minimum perturbation norm. Here, we use the average gradient of all exits in the multi-exit neural network as the gradient for updating adversarial examples. Similar to above, we label a sample as adversarial if it can mislead all exits in the multi-exit neural network.

### H.2.3 Square

The Square attack generates adversarial attacks via random searches of adversarial patches with variable degrees of perturbation and position. The Square attack algorithm iteratively samples the perturbation degree and the positions of patches while reducing the size of patches. The sampled adversarial patches are added to a sample, and the patches that maximizes the loss of the target model are selected. Here, we use the average loss of all exits in the multi-exit neural network as the loss for determining whether a sampled perturbation increases or decreases the loss of the target model. Additionally, we determine a sample as adversarial if it misleads all exits in the multi-exit neural network.

## H.3 Experiment details

For all attacks, we commonly use $\epsilon = 0.03$ as the perturbation degree and generate adversarial examples over 50 steps. All the attacks are based on $L_\infty$ norm. For the APGD attack, we employ cross entropy loss for computing the gradient to update adversarial examples. In both the APGD-T and FAB attacks, all class pairs are considered when generating adversarial attacks. For the Square attack, the random search operation is conducted 5000 times (the number of queries). Other settings follow [14]. In terms of performance comparison against stronger attacker algorithms, we adopt adversarial training via average attack in both Adv. w/o Distill [12] and NEO-KD (our approach). However, since the original CW attack algorithm and AutoAttack algorithm were designed for single-exit neural networks, this adapted versions targeting multi-exit neural networks are relatively weak.

