# OpenReview forum: "NEO-KD: Knowledge-Distillation-Based Adversarial Training for Robust Multi-Exit Neural Networks"
_NeurIPS.cc/2023/Conference — NeurIPS 2023 poster_

### Official Review · Reviewer_grwc · 2023-07-04

**Soundness:** 2 fair
**Presentation:** 3 good
**Contribution:** 2 fair
**Rating:** 4
**Confidence:** 4

**Summary:**

This paper proposes a knowledge distillation approach for defending against adversarial attacks in multi-exit networks. They have two different objectives: (i) leveraging self-distillation to improve adversarial robustness, (ii) reducing adversarial transferability among the submodules of the network. Since multi-exit networks have submodules that are correlated with each other, distilling knowledge from the output of clean data at the last exit to all submodules  increases the chances of adversarial transferability. To avoid this, this paper proposes a two-fold approach: (i) Neighbor knowledge distillation (NKD): generates teacher prediction at exit i by averaging the predictions of clean data from exits i-1, i and i+1 and distills it to adversarial predictions at exit i; (ii) Exit-wise orthogonal knowledge distillation (EOKD): output of clean data at exit i is distilled to adversarial example at exit i. Orthogonal labeling operation on clean predictions makes teacher predictions orthogonal across all exits. NKD helps create high quality teacher predictions improving adversarial robustness of multi-exit networks. At the same time, NKD creates different teacher predictions for each exit reducing the risk of adversarial transferability. EOKD further reduces chances of adversarial transferability.

**Strengths:**

1. The paper is well-written. The approach is well explained and the claims have been substantiated with results.

2. NEO-KD achieves best adversarial accuracy against max-average and average attacks in all budget setups.

3. The paper successfully demonstrates reduction in adversarial transferability among exits in multi-exit networks

**Weaknesses:**

1. In anytime prediction setup, NEO-KD shows low performance at the later exits for small datasets like MNIST, Cifar-10

2. NEO-KD shows low performance at early exit for mid scale datasets like TinyImageNet.

**Questions:**

Please demonstrate results with ViT-tiny or small.

Please use large scale dataset like ImageNet to show the efficacy of the method.

**Limitations:**

Large scale demonstration missing, given that it is an empirical paper.

---

> ### Author Rebuttal · Authors · 2023-08-09
>
> We appreciate the reviewer for the time and efforts, and providing  helpful comments that are also very clear. Below, we provide responses to the reviewer's comments.
>
> ### **NEO-KD shows low performance at the specific exits in some cases.**
>
> We agree with the reviewer that in some datasets, NEO-KD occasionally does not achieve the best performance in specific exits. However, we respectfilly disagree that NEO-KD achieves low performance:  NEO-KD performs the best in all 7 exits for CIFAR-100 considering top-1 accuracy, performs the best in 4 out of 5 exits on Tiny-ImageNet. For small datasets (e.g., CIFAR-10), NEO-KD only looses against the baseline in 0 or 1 exit. Even in these exits, NEO-KD still achieves the second best performance most of the time, where the gap with  the best performance is marginal. NEO-KD also achieves the best average accuracy in all cases. Hence, when a system designer has  to decide which algorithm  to use for constructing a robust multi-exit neural network, it is easy to answer that the proposed NEO-KD is the best.  Overall, we believe that this strength of the proposed scheme as well as the fact that NEO-KD is the very first work to strategically integrate (i) _multi-exit networks_, (ii) _self-distillation_, and (iii) _adversarial training_, deserve merits in both multi-exit networks and adversarial training communities.
>
> ### **Additional experiments.**
>
> We appreciate the suggestion. To comply with the reviewer's comment, we conducted additional experiments using ImageNet with 1000 object classes. Due to the strict timeline, we considered ImageNet(100) where 100 samples are selected from each class in the ImageNet to construct the training set. Considering that there are 1000   classes in ImageNet, the number of train samples we use is 100,000. This dataset has been adopted in many multi-exit network literature to prove the effectiveness of the algorithm [ICCV'19], [AAAI'21]. The results in the below table show that the proposed NEO-KD performs the best in the larger-scale dataset with 1000 classes, further confirming its advantage.
>
> | Exit | 1 | 2 | 3 | 4 | 5 | Average |
> |:---|:---:|:---:|:---:|:---:|:---:|:---:|
> | Adv. w/o Distill | 18.54% | 24.66% | 27.63% | 28.28% | 29.71% | 25.76% |
> | NEO-KD (ours) | 21.86% | 28.04% | 31.15% | 31.93% | 33.87% | 29.37% |
>
> We also appreciate the reviewer for suggesting experiments with ViT models. However, due to strict timeline for  implementing and training  the baselines as well as our scheme with ViT models, we were not able to provide the corresponding results.  We agree that  there  exists some works that aim to combine multi-exit networks and  transformers, which itself is a challenging research topic   in the multi-exit network  community. However,  since our solution integrates   multi-exit network, self-distillation,  and adversarial training altogether (which has not been considered in the literature), adding another transformer-side dimension beyond these components   makes the problem setup extremely complicated and  causes  significant resource issues, compared to the case with only combining transformer with multi-exit network (with no adversarial training and no self-distillation).  Please  also consider that we  already used the most  commonly adopted model in multi-exit network literature  i.e., MSDNet.
>
> Overall, we   have now considered   5 datasets -- MNIST, CIFAR-10, CIFAR-100, Tiny-ImageNet, and ImageNet -- and trained SmallCNN and 3 different variations of MSDNet (with 3, 5, 7 exits).
>
> [ICCV'19] Phuong et al., ``Distillation-Based Training for Multi-Exit Architectures,'' ICCV 2019.
>
> [AAAI'21] Wang et al., ``Harmonized Dense Knowledge Distillation Training for Multi-Exit Architectures,'' AAAI 2021.
>
> Again, thank you for your time and efforts, providing helpful comments to improve our paper. We would appreciate further opportunities to answer any remaining concerns you might have.

---

> ### Comment · Senior_Area_Chairs · 2023-08-21
> **final discussions**
>
> Dear Reviewer,
>
> As discussions come to an end soon, this is a polite reminder to engage with the authors in discussion.
> Please note we take note of unresponsive reviewers.
>
> Best regards,
> \
> SAC

---

> ### Comment · Reviewer_grwc · 2023-08-21
> **Thanks**
>
> Thanks to the authors for their detailed response. I have one follow-up question, is the method applicable in case of a hybrid distillation framework like as shown in [1]. I would encourage the authors to provide some insight and possibly results on that.
>
> [1] Analyzing the confidentiality of undistilable teachers in knowledge distillation, NeurIPS 2021.

---

> > ### Author Response · Authors · 2023-08-21
> >
> > Thanks for the response. The paper you suggested propose a hybrid distillation methodology as follows: (i) First, a skeptical student strategy is proposed to adopt intermediate shallow classifier: this prevents the information leakage from the teacher to the student. (ii) Secondly, the student adopts self distillation to improve the learnability of the student.
> >
> > We need to highlight that our scheme is a self-distillation-based strategy that does not have an external teacher network. The main goal of [1] is to prevent the model stealer (i.e., student) to extract the knowledge of the teacher: the technical details of [1] were developed to achieve this. However, since everything is conducted by a single multi-exit network in our case based on the self-distillation during adversarial training, we actually do not need to worry about the data leakage of the teacher to another student.

---

> > > ### Comment · Reviewer_grwc · 2023-08-21
> > > **Re**
> > >
> > > In my comment, I requested for the applicability of your method under situation when we do distill from an external teacher as well, added to self distillation. Thanks for nice summary of the paper [1], though I am aware of it. Please understand that I do not expect your paper to be discussed in the context of data leakage or model IP, like [1] did, I just want to get clarification on the possible applicability under hybrid (both self and external teacher) distillation.

---

> > > > ### Author Response · Authors · 2023-08-21
> > > >
> > > > Thank you for the prompt response and clarification. Now we understand your question better.
> > > >
> > > > Suppose we have an additional teacher network, and this teacher network wants to improve the performance of our multi-exit network which will be trained based on self-distillation. Of course in this scenario, our scheme can work as a hybrid: while our scheme is conducting self-distillation based on NKD and EOKD, the additional knowledge of the teacher network can be distilled to the exits of our student network. Here, we would like to provide the similar insight that we explained in the main manuscript: If we distill the same knowledge of the external teacher network to all the exits of our student model, the adversarial transferability can get increased due to the increased dependencies among exits (or submodels) in the student network. Hence, similar techniques adopted in our work (e.g., orthogonal label distillation) should be utilized when the teacher distills the knowledge to our student network, which will eventually reduce the adversarial transferability to gain robustness against adversarial attacks in multi-exit networks (which is our focus).
> > > >
> > > > Hope this could provide insights into the applicability of our work to hybrid distillation.
> > > >
> > > > Thanks again for your insightful comments.
> > > >
> > > > Best,
> > > >  Authors

---

> > > > > ### Author Response · Authors · 2023-08-22
> > > > >
> > > > > Dear Reviewer grwc, we are contacting again as we are getting a bit anxious as the end nears. Would our answer be satisfactory? Please let us know if there is anything we can do to help clarify any remaining issues.

---

### Official Review · Reviewer_KNG2 · 2023-07-06

**Soundness:** 4 excellent
**Presentation:** 4 excellent
**Contribution:** 4 excellent
**Rating:** 6
**Confidence:** 5

**Summary:**

This paper proposed a know-distillation-based method to improve the adversarial robustness of multi-exit neural networks. Extensive experiments are conducted to show the effectiveness of the proposed method.

**Strengths:**

1. Extensive experiments are conducted to show the effectiveness of the proposed method.
2. This method is a novel combination of previous methods.
3. The paper is well-organized and easy to follow.
4. Technical details are provided in the supplementary materials.
5. Ablation study is provided.

**Weaknesses:**

1. This method is to improve the adversarial robustness of DNNs, but this paper lacks a comparison with some general defense methods, e.g., TEAT[1].
2. What is the advantage of distillation methods over other adversarial defense methods on this task?
3. "One challenge is that simply applying existing self-distillation techniques increases the adversarial transferability across different submodels, since the same knowledge from the last exit is distilled to all the other exits, increasing the dependency among different submodels in the network. "
I can find any support for this claim in this paper.
4. What is the difference between the proposed method and other related work? This paper lacks a detailed/formal comparison with other related methods.

Dong, Yinpeng, et al. "Exploring memorization in adversarial training." ICLR 2022

**Questions:**

1. "Considering our distillation method, NEO-KD is currently not directly applicable to object detection."
Can the author briefly explain why it cannot be applied to object detection?


**Limitations:**

The limitations are clearly stated.

---

> ### Author Rebuttal · Authors · 2023-08-09
>
> We appreciate the reviewer for the positive comments and valuable feedback. Below, we provide answers to the comments raised by the reviewer.
>
> ### **Comparison with TEAT**
> The baselines in the main paper were generally the adversarial defense methods designed for multi-exit network. As the reviewer suggested, we conducted additional experiments with TEAT [ICLR'22] and compare with our method. Since TEAT was originally designed for the single-exit network, we first adapted TEAT to the multi-exit network setting. Instead of the original TEAT that generates the adversarial examples considering the final output of the network, we modified TEAT to generate adversarial examples that maximizes the average loss of all exits in the multi-exit network. Table R5 above shows the results using max-average attack on CIFAR-10/100 dataset. It can be seen that our NEO-KD, which is designed for multi-exit network, achieves higher adversarial test accuracy compared to the TEAT methods (PGD-TE and TRADES-TE) designed for single-exit networks. The results highlights the necessity of developing adversarial defense techniques geared to multi-exit networks rather than adapting general defense methods  used for single-exit network.
>
> ### **Advantages of distillation-based approaches over adversarial defense methods**
> Our distillation-based approach for robust multi-exit networks has the following key advantages over conventional adversarial defense methods:
>
> First, it directly tackles the unique challenges that arise in multi-exit networks. Multi-exit networks are highly vulnerable to simple attacks (e.g., an attack targeting a single exit) since it has high dependencies between exits. Existing adversarial defense approaches cannot directly handle this unique problem of multi-exit network, as they focus on the single-exit network. Even when these methods are adapted to multi-exit networks, they face the same challenges as can be also seen in our results with TEAT above. On the other hand, distillation-based methods provide a great platform to tackle this issue, as each exit can be distilled with different knowledge to reduce the dependencies among exits. Our NEO-KD strategically utilizes distillation based on neighbor ensembling, exit-wise, and orthogonal distillation.
>
> Another advantage of the distillation method in defending against adversarial attacks is its compatibility with other defense strategies. Specifically, our distillation-based scheme is an orthogonal approach to existing defense methods, as the proposed distillation losses can be used with any classification loss (including adversarial training loss). As can be seen from experimental results of the main paper, our NEO-KD combined with existing adversarial loss (via max-average attack [ICLR'20]) brings large performance gains, showing its good compatibility.
>
> ### **Do the prior self-distillation methods really increase the adversarial transferability?**
> This claim can be supported by the results in Fig. 4 of our main paper. The average adversarial transferability of SKD (which adopts self-distillation) is 33.36%, which is significantly larger than the transferability of the baseline without distillation, which is 23.68%. This makes sense since the same knowledge from the last exit is distilled to all other exits, which results in increased dependencies among all submodels. On the other hand, our NEO-KD achieves the lowest adversarial transferability (20.12%) based on the proposed neighbor distillation and exit-wise orthogonal distillation, that are designed to reduce the transferability among submodels. This advantage of NEO-KD in adversarial transferability contributes to a better adversarial test accuracy compared to the baselines, as shown in the main paper.
>
> ### **More detailed comparison with related works**
> Since this paper focuses on the robustness of multi-exit neural networks, we mainly compared our NEO-KD approach with other self-distillation and defense methods used in the context of multi-exit networks, as discussed in the related work section (Section 2) of the main paper. Below, we provide a more detailed comparison with existing works relevant to conventional adversarial training.
>
> In the context of defense schemes against adversarial examples, conventional adversarial training methods [ICML'19, ICLR'22, CVPR'23] proposed for single-exit network have mainly focused on creating new adversarial training losses restricted to single-exit network. Therefore, adversarial transferability between exits has not been a key issue in the prior works for single-exit network. Recent methods targeting multi-exit network [ICLR'20] have also proposed new adversarial training losses, but they still do not directly handle the adversarial transferability, which is a inherent problem in multi-exit networks. Our NEO-KD is an orthogonal approach focusing on directly mitigating adversarial transferability across exits and can be combined with the existing adversarial training losses.
>
> In summary, our work provides distinct advantages compared to prior works by focusing on mitigating adversarial transferability which is a unique challenge of multi-exit network, improving robustness of multi-exit network.
>
> ### **Application to object detection task**
> When applying distillation to object detection in multi-exit network, we need to distill predictions of box regression and classification from the teacher (last exit) to the students (early exits). However, since the box regression predictions can differ between the teacher and the students, the student may fail to detect an object box that the teacher can identify, making it difficult to apply to object detection. We believe our work could pave the way to develop distillation-based adversarial training for multi-exit network in more complicated tasks, such as object detection.
>
> Again, thank you for the positive and helpful comments. We will make all of these points clearer in the revised manuscript.

---

### Official Review · Reviewer_PqqG · 2023-07-06

**Soundness:** 3 good
**Presentation:** 3 good
**Contribution:** 2 fair
**Rating:** 4
**Confidence:** 4

**Summary:**

This paper proposed a knowledge-distillation based adversarial training method, which is designed for multi-exit neural networks. The authors propose neighbor knowledge distillation to improve the robustness against adversarial attacks, and propose exit-wise orthogonal knowledge distillation to reduce the adversarial transferability across different submodels. Moreover, the proposed method is a plug-and-play method, which can be used in prevailing training strategies of multi-exit networks.

**Strengths:**

1.	This paper brings self-distillation into adversarial training, and make a good combination with multi-exit networks.
2.	The experimental results on several datasets and attacks demonstrate its superiority, and the considered scenarios are sufficient.


**Weaknesses:**

1.	Though the experimental result is extraordinary, the motivation and operation of EOKD makes me confused. I did not really get the meaning of orthogonal labeling operation $O(\cdot)$, and the paper does not clearly elaborate the relation between orthogonal labeling and adversarial transferability reduction. Moreover, I would like to know whether EOKD will change the behavior of the output from every exit on clean examples.
2.	The settings of baselines need to be discussed. The author conduct experiments of distillation-based baselines (SKD and ARD) by utilizing the prediction of the last exit, instead of every exit or random exit or another specific exit. Is there any reason to choose the last exit in baseline methods?


**Questions:**

1.	I hope the authors can clarify the motivation of EOKD and the orthogonal labeling operation more clearly.
2.	Please explain the setting of baselines which only use the prediction of last exit to conduct knowledge distillation.
3.	I would like to know the training speed of the proposed NEO-KD, as there are many extra self-distillation terms, which will complicate the computational graph.


**Limitations:**

As a new adversarial training method, this paper builds a comprehensive solution, and I think this paper have discussed their limitations enough.

---

> ### Author Rebuttal · Authors · 2023-08-09
>
> We appreciate the reviewer for the helpful comments, especially on the unclear aspects in the paper. In the response below, we would like to clarify all the ambiguous points raised by the reviewer.
>
> ### **Motivation of EOKD**
> Multi-exit networks are highly vulnerable to simple attacks (e.g., an adversarial attack targeting a single exit) as the submodels in the network are highly correlated by sharing some model parameters. The motivation of EOKD is to _reduce the dependencies among exits_ while taking advantage of knowledge-distillation during adversarial training of multi-exit networks. To achieve this, EOKD is equipped with two key components: (i) exit-wise distillation and (ii) orthogonal distillation. The idea of the first component is to distill the knowledge of clean data to the output of the adversarial sample in an exit-wise manner, which intuitively reduces the interdependencies between exits by distilling a different knowledge to each exit (this has a clear difference compared to prior self-distillation strategies [ICCV'19a], [ICCV'19b] that distill the same knowledge to all exits). The second component of EOKD is based on the orthogonal labeling operation $O(\cdot)$ to encourage the predictions for the non-ground-truth classes of individual exits to be mutually orthogonal, by providing orthogonal soft labels.
>
> ### **Motivation of orthogonal labeling and its operation.**
> The EOKD loss function is defined as $EOKD_{i,j}=\ell(f_{\theta_i}(x^{adv}_j), O(f _{\theta_i}(x_j)))$ as in Eq. (6) of our main paper. To see how the orthogonal labeling operation $O(\cdot)$ works, consider a toy example with 3-exit network (i.e., $L=3$) focusing on a 4-way classification task (i.e., $C=4$). Let  $[p^i_1, p^i_2, p^i_3, p^i_4]$ be the softmax output of the clean sample at the $i$-th exit, for $i=1,2,3$. If class 1 is the ground-truth, the orthogonal labeling operation $O(\cdot)$ jointly produces the following results from each exit: $[\hat{p}^1_1, \hat{p}^1_2, 0, 0]$ from exit 1, $[\hat{p}^2_1, 0, \hat{p}^2_3, 0]$ from exit 2, $[\hat{p}^3_1, 0, 0, \hat{p}^3_4]$ from exit 3, where $\hat{p}$ indicates the normalized probability of $p$ so that the values in each vector sum to one. Here, it can be seen that except the prediction $\hat{p}_1^i$ for class 1, the non-ground-truth predictions become orthogonal across different exits with no overlappings. This strategy enables the model to distill the ground-truth information to all exits while maximizing the distinction of distilled knowledge for each exit, reducing the dependencies among exits (and thus improving adversarial transferability).
>
> More generally speaking, for each exit, $O(\cdot)$ randomly selects $\lfloor (C-1)/L \rfloor$ labels among total of $C$ classes so that   the selected labels are non-overlapping across different exits (except for the answer label), where the probabilities of selected labels are normalized to sum to one. This makes the predictions  for the non-ground-truth classes to be orthogonal across all exits. By doing so, the essential knowledge of the ground-truth class is preserved while the knowledge of other classes is orthogonally distilled, promoting the reduction of dependencies between exits. In Fig. 4 of the main paper, it can be seen that EOKD can effectively reduce the adversarial transferability. This also leads to an improved adversarial test accuracy compared with the baselines, as can be seen in Tables 1,2,3,4 of our main manuscript.
>
> ### **Effect of EOKD on clean examples**
> Table R1 & R2 above shows the comparison between NKD and NEOKD in terms of clean/adversarial test accuracy using CIFAR-10. As can seen from the results, it is confirmed that applying EOKD slightly compromises the clean accuracy but yields large performance gain in adversarial test accuracy.
>
> ### **Why do baselines (SKD, ARD) use the last exit for distillation?**
> We would like to first clarify the reason why the last exit is utilized in implementing the baselines (SKD [ICCV'19a], ARD [AAAI'20]), and then provide additional experimental results using different distillation strategies for the baselines (e.g., using another exit or ensemble of exits instead of the last one). Existing self-distillation schemes [ICCV'19a, ICCV'19b] for multi-exit network improve the performance on clean samples by self-distilling the knowledge of the last exit, as the last exit has the best prediction quality. Therefore, following the original philosophy, we also used the last exit in implementing the SKD baseline. Regarding ARD [AAAI'20], since it was proposed for single-exit network, we also utilized the last exit with high performance when applying ARD to multi-exit networks. Nevertheless, as per the reviewer's suggestion, we performed additional experiments to consider comprehensive baselines using various exits for distillation. Table R3 above shows the results of SKD and ARD using a specific exit or an ensemble of all exits for distillation. The results show that our scheme consistently outperforms all of the baselines.
>
> ### **Training speed**
> In Table R4, we compare the training time of our scheme and the considered baselines for one epoch on CIFAR-100 dataset. It is observed that our scheme requires 4\% more time than the basic adversarial training (i.e., Adv. w/o Distill), and only 1.6\% more time than other KD based baselines (SKD, ARD). Considering that the main focus of multi-exit networks is the latency during inference rather than training, this small additional computation during training can be seen as a reasonable cost for achieving an improved adversarial test accuracy compared to the baselines.
>
> Again, thank you for your time and efforts in reviewing our paper. Your raised concerns made us think deep and wide; and we feel we have managed to clarify all the issues raised. We would appreciate further opportunities to answer any remaining concerns you might have.

---

> ### Comment · Senior_Area_Chairs · 2023-08-21
> **final discussions**
>
> Dear Reviewer,
>
> As discussions come to an end soon, this is a polite reminder to engage with the authors in discussion.
> Please note we take note of unresponsive reviewers.
>
> Best regards,
> \
> SAC

---

> ### Comment · Reviewer_PqqG · 2023-08-21
>
> The reviewer would like to appreciate the responses from the authors. However, the motivation is still somewhat confusing to me and I incline to keep my original score.

---

> > ### Author Response · Authors · 2023-08-21
> >
> > Dear Reviewer PqqG
> >
> > Thanks for your response, but we would be grateful if you could be a bit more specific on which part of the motivation is still confusing. In the above response, we have tried to clearly illustrate the motivations of EOKD and the orthogonal labeling process in detail using a simple **toy example with a 3-exit network and a 4-way classification task**, which is also supported by the experiments in the main manuscript, and we honestly feel that there is nothing we can add further to make it clearer.
> >
> > We had a discussion period of around two weeks, and we must say it feels unfair that the reviewer didn't provide any comment until the discussion period is 6 hours left (with regional time differences); we were disappointed especially because the reviewer's only remaining concern is clarification on motivation, which is generally easy to address.
> >
> > Again, we would be grateful if you could let us know which part of the motivation is unclear to you.
> >
> > Best, Authors

---

### Official Review · Reviewer_iEEe · 2023-07-24

**Soundness:** 2 fair
**Presentation:** 2 fair
**Contribution:** 2 fair
**Rating:** 6
**Confidence:** 3

**Summary:**

The paper presents a novel method called Neighbor Exitwise Orthogonal Knowledge Distillation (NEO-KD) for improving the adversarial robustness of multi-exit networks. The method's motivation lies in addressing the issue that existing knowledge distillation schemes are not ideal for multi-exit networks as they either increase or decrease adversarial robustness. The authors argue that the choice of the knowledge to distill and the specific exit to target significantly influence the robustness of multi-exit networks.

- NKD improves the network's defense by ensuring the outputs of adversarial examples mimic the outputs of clean data, by distilling the combined predictions of clean data neighbor exits, leading to enhanced robustness and superior feature quality for corresponding exits.

- EOKD concentrates on minimizing adversarial transferability between different network submodels, by distilling the output of clean data to the output of adversarial samples in an exit-by-exit manner, while promoting orthogonality in non-maximal predictions of individual exits.

- Experiments demonstrate that NEO-KD outperforms existing solutions across a range of commonly adopted datasets, including MNIST, CIFAR-10/100, and Tiny-ImageNet.
- NEO-KD demonstrates superior performance under different prediction setups (anytime and budgeted), and exhibits reduced adversarial transferability.
- Ablation studies demonstrated the benefits of its individual components and its robustness against stronger adversarial attacks.
Overall, the paper offers a substantial contribution to improving the adversarial robustness of multi-exit networks.

**Strengths:**

1) The paper presents a novel approach called NEO-KD that uses Neighbor Knowledge Distillation (NKD) and Exitwise Orthogonal Knowledge Distillation (EOKD) for enhancing adversarial robustness in multi-exit networks. This approach is distinct in its application of knowledge distillation techniques to the unique challenges posed by multi-exit networks.

2) An extensive set of experiments were conducted to validate the proposed method using different datasets and adversarial attacks. This includes the use of anytime and budgeted prediction setups, and analysis of adversarial transferability. The results show the NEO-KD method's performance relative to baseline techniques.

3) The paper(motivation) addresses an important problem in the area of multi-exit networks: how to enhance adversarial robustness. The proposed solution, NEO-KD, not only improves robustness, but also optimizes computation costs and reduces adversarial transferability.

**Weaknesses:**

The approach is tested on four standard datasets: MNIST, CIFAR-10, CIFAR-100, and Tiny-ImageNet. While these are common benchmarks, the effectiveness of NEO-KD on larger datasets such as ImageNet is not demonstrated.

**Questions:**

The paper incorporates the application of ensemble strategy at inference time in the budgeted prediction setup. It would be interesting to learn more about your decision process for using this approach. Specifically, how did you determine the confidence threshold for selecting the exit? Are there any trade-offs or implications if the threshold was set differently?

**Limitations:**

- The experiments were performed using relatively smaller datasets (CIFAR-10, CIFAR-100, MNIST, and Tiny-ImageNet), and the performance on larger datasets such as ImageNet remains unexplored.
- The choice and implications of the confidence threshold used in the ensemble strategy during the budgeted prediction setup are not clearly explained.

---

> ### Author Rebuttal · Authors · 2023-08-09
>
> We appreciate the reviewer for the time and efforts. In the response below, we provide answers to the comments raised by the reviewer.
>
> ### **Results on larger datasets**
>
> We appreciate the suggestion. We conducted additional experiments using ImageNet with 1000 object classes. Due to the strict timeline, we considered ImageNet(100) where 100 samples are selected from each class in the ImageNet to construct the training set. Considering that there are 1000   classes in ImageNet, the number of train samples we use is 100,000. This dataset has been adopted in many multi-exit network literature to prove the effectiveness of the algorithm [ICCV'19], [AAAI'21]. The results in the below table show that the proposed NEO-KD performs the best in the larger-scale dataset with 1000 classes, further confirming its advantage.
>
> | Exit | 1 | 2 | 3 | 4 | 5 | Average |
> |:---|:---:|:---:|:---:|:---:|:---:|:---:|
> | Adv. w/o Distill | 18.54% | 24.66% | 27.63% | 28.28% | 29.71% | 25.76% |
> | NEO-KD (ours) | 21.86% | 28.04% | 31.15% | 31.93% | 33.87% | 29.37% |
>
> ### **Accuracy-computation trade-off controlled by the confidence threshold**
>
> In a budgeted prediction setup, given a limited computing budget, the trained model has to make predictions for all samples within the budget. For instance, if the budget is sufficiently provided, we can obtain high performance by classifying many samples at the later exits. On the other hand, if the budget is significantly small, the model should classify most of the samples at early exits while compromising performance. In other words, the model should make predictions efficiently within limited budget constraints by classifying 'easy' samples at early exits and 'hard' samples at later exits. At this time, _confidence threshold_, which is the maximum value of the prediction probability vector by softmax, is a criteria for determining whether to perform prediction on a sample at the current exit or the next exit. Note that each exit has a different confidence threshold, which is determined using validation set (this will be clarified  soon). Specifically, given a sample, if the confidence of the sample at a specific exit is greater than the predefined threshold at the exit (easy sample), the prediction is made on the current exit, and if it is lower than the threshold (hard sample), the feature of the sample is passed to the next exit for prediction at the next exit. Therefore, confidence threshold controls the trade-off between computational cost and accuracy. As the threshold increases, more samples are predicted at the earlier exits, reducing the number of samples forwarded to the later exits.  Consequently, fewer resources are used for classifying test samples while compromising the performance. Conversely, when the threshold is decreased, more samples are predicted at later exit, which consumes more resources and yields higher performance.
>
> ### **How to determine confidence threshold**
>
> As per the reviewer's suggestion, we would like to provide a detailed explanation about how to determine confidence threshold for each exit using validation set before the testing phase. First, in order to obtain confidence thresholds for various budget scenarios, we allocate the number of validation samples for each exit. For simplicity, consider a toy example with 3-exit network (i.e., $L = 3$) and assume the number of validation set is 3000. Then, each exit can be assigned a different number of samples: for instance, (2000, 500, 500), (1000, 1000, 1000) and (500, 1000, 1500). As more samples are allocated to the early exits, a scenario with a smaller budget can be obtained, while allocating more data to the later exits can lead to a scenario with a larger budget. More specifically, to see how to obtain the confidence threshold for each exit, consider the low-budget case of (2000, 500, 500). The model first makes predictions on all 3000 samples at exit 1 and sorts the samples based on their confidence. Then, the 2000th largest confidence value is set as the confidence threshold for the exit 1. Likewise, the model performs predictions on remaining 1000 samples at exit 2 and the 500th largest confidence is determined as the threshold for the exit 2. Following this process, all thresholds for each exit are determined. During the testing phase, we perform predictions on test samples based on the predefined thresholds for each exit, and calculate the total computational budget for the combination of (2000, 500, 500). In this way, we can obtain accuracy and computational budget for different combinations of data numbers (i.e., various budget scenarios). Fig. 2 and 3 in the main paper show the results for 100 cases of different budget scenarios.
>
> Again, we appreciate the reviewer for the insightful comments. We would be happy to address any remaining questions the reviewer might have.
>
> [ICCV'19] Phuong et al., ``Distillation-Based Training for Multi-Exit Architectures,'' ICCV 2019.
>
> [AAAI'21] Wang et al., ``Harmonized Dense Knowledge Distillation Training for Multi-Exit Architectures,'' AAAI 2021.
>
> Again, we appreciate the reviewer for the positive comments with valuable feedback. We would appreciate further opportunities to answer any remaining concerns you might have.

---

> ### Comment · Reviewer_iEEe · 2023-08-20
> **Post-Rebuttal Comment:**
>
> Based on the detailed response from the authors addressing my concerns and clarifications, and taking into consideration the feedback from other reviewers and the rebuttal: I'm glad to see the added tests on ImageNet(100) that tackle my worries about scalability. Their clarification on the confidence threshold in the budgeted prediction setup clears up my earlier questions.
>
> Seeing how the authors addressed the feedback and the paper's importance for multi-exit networks, I'm leaning towards changing my rating from 'Borderline accept' to 'Weak Accept'.

---

> > ### Author Response · Authors · 2023-08-20
> >
> > Thank you very much for acknowledging our efforts and raising your score.
> >
> > Best, Authors of Paper 11001

---

### Author Rebuttal · Authors · 2023-08-09

We appreciate all reviewers for providing constructive comments, which have greatly helped us to improve the paper.

Due to the limited content we can provide in each response, we would like to share additional experimental results that **Reviewer PqqG** and  **Reviewer KNG2** suggested here. For the other reviewers (Reviewer iEEe and Reviewer grwc), all results are provided in our response corresponding to each reviewer.

In general, all approaches, including the baselines and our NEO-KD, are trained using adversarial examples generated through the max-average attack.


&nbsp;

### Tables \& References for **Reviewer PqqG**



* **Table R1:** Clean test accuracy of NKD and NEO-KD.
|Exit|1|2|3|Average Clean Accuracy|
|:---|:---:|:---:|:---:|:---:|
|NKD|76.81%|79.03%|81.98%|79.27%|
|NEO-KD|75.77%|78.37%|81.37%|78.50%|

* **Table R2:** Adversarial test accuracy of NKD and NEO-KD.
|Exit|1|2|3|Average Adversarial Accuracy|
|:---|:---:|:---:|:---:|:---:|
|NKD|46.48%|46.63%|44.64%|45.92%|
|NEO-KD|46.53%|47.65%|50.71%|48.30%|


* **Table R3:** Adversarial test accuracy of SKD and ARD according to exit selection as a teacher prediction.
|Exit|1|2|3|Average|
|:---|:---:|:---:|:---:|:---:|
|SKD (exit 1)|32.27%|36.92%|38.57%|35.92%|
|SKD (exit 2)|35.33%|35.10%|37.82%|36.08%|
|SKD (exit 3)|39.36%|41.39%|38.39%|39.71%|
|SKD (ensemble)|38.63%|41.80%|40.13%|40.19%|
|ARD (exit 1)|35.64%|38.10%|42.12%|38.62%|
|ARD (exit 2)|35.35%|38.24%|40.00%|37.86%|
|ARD (exit 3)|39.37%|41.98%|43.53%|41.63%|
|ARD (ensemble)|35.22%|38.35%|40.76%|38.11%|
|NEO-KD (ours)|41.67%|45.38%|45.54%|44.20%|


* **Table R4:** Comparison of training time between our NEO-KD and baselines.
|Method|Adv. w/o Distill|SKD|ARD|NEO-KD|
|:---|:---:|:---:|:---:|:---:|
|Training time (min/epoch) |13.40|13.80|13.80|14.02|



[ICCV'19a] Phuong et al., Distillation-based training for multi-exit architectures.

[ICCV'19b] Li et al., Improved techniques for training adaptive deep networks.

[AAAI'20] Goldblum et al,. Adversarially robust distillation.





&nbsp;


### Tables \& References for **Reviewer KNG2**

**Table R5:** Comparison of adversarial test accuracy against max-average attack between TEAT methods and our NEO-KD.

* (a) CIFAR-10
|Exit|1|2|3|Average|
|:---|:---:|:---:|:---:|:---:|
|PGD-TE [ICLR'22]|**48.73%**|46.00%|46.85%|47.19%|
|TRADES-TE [ICLR'22]|45.05%|39.64%|42.10%|42.26%|
|NEO-KD (ours)|46.53%|**47.65%**|**50.71%**|**48.30%**|

* (b) CIFAR-100
|Exit|1|2|3|4|5|6|7|Average|
|:---|:---:|:---:|:---:|:---:|:---:|:---:|:---:|:---:|
|PGD-TE [ICLR'22]|24.07%|24.39%|25.14%|25.35%|26.29%|25.57%|24.60%|25.06%|
|TRADES-TE [ICLR'22]|17.62%|18.52%|18.61%|18.98%|18.95%|19.67%|20.35%|18.96%|
|NEO-KD (ours)|**28.37%**|**28.78%**|**29.02%**|**29.49%**|**30.06%**|**28.45%**|**28.54%**|**28.96%**|


[ICLR'22] Dong, Yinpeng, et al. ``Exploring memorization in adversarial training,'' ICLR 2022.

[ICLR'20] Hu et al., Triple wins: Boosting accuracy, robustness and efficiency together by enabling input-adaptive inference.

[ICML'19] Zhang et al. Theoretically principled trade-off between robustness and accuracy.

[ICLR'22] Dong et al. Exploring memorization in adversarial training.

[CVPR'23] Dong et al., The Enemy of My Enemy is My Friend: Exploring Inverse Adversaries for Improving Adversarial Training.

[ICLR'20] Hu et al., Triple wins: Boosting accuracy, robustness and efficiency together by enabling input-adaptive inference.

[ICCV'19a] Phuong et al., Distillation-based training for multi-exit architectures.

[ICCV'19b] Li et al., Improved techniques for training adaptive deep networks.

---

### Decision · Program_Chairs · 2023-09-21

**Decision:**

Accept (poster)

**Comment:**

The paper discusses an adversarial training approach based on knowledge distillation for multi-exit networks. The initial reviews raised questions regarding experiments on larger datasets, the role of hyper-parameters such as the confidence threshold,  motivation of components such as EOKD, training time, clarification on baseline, and comparison with TEAT. The AC feels the combination of adversarial training with self-distillation in the context of multi-exit networks is interesting and novel and the results are encouraging. Therefore, the AC recommends acceptance and would remind authors to include promised details and results in the revised version.